# Multimodal measures of spontaneous brain activity reveal both common and divergent patterns of cortical functional organization

Hadi Vafaii [1] ✉, Francesca Mandino [2], Gabriel Desrosiers-Grégoire [3,4], David O'Connor [5], Marija Markicevic[2], Xilin Shen[2], Xinxin Ge [6], Peter Herman [2], Fahmeed Hyder [2], Xenophon Papademetris[2,5,7], Mallar Chakravarty[3,4,8,9], Michael C. Crair [10,11,12], R. Todd Constable [2,5,13], Evelyn M. R. Lake [2,5,17] ✉ & Luiz Pessoa [14,15,16,17] ✉

Large-scale functional networks have been characterized in both rodent and human brains, typically by analyzing fMRI-BOLD signals. However, the relationship between fMRI-BOLD and underlying neural activity is complex and incompletely understood, which poses challenges to interpreting network organization obtained using this technique. Additionally, most work has assumed a disjoint functional network organization (i.e., brain regions belong to one and only one network). Here, we employ wide-field $Ca^{2+}$ imaging simultaneously with fMRI-BOLD in mice expressing GCaMP6f in excitatory neurons. We determine cortical networks discovered by each modality using a mixed-membership algorithm to test the hypothesis that functional networks exhibit overlapping organization. We find that there is considerable network overlap (both modalities) in addition to disjoint organization. Our results show that multiple BOLD networks are detected via $Ca^{2+}$ signals, and networks determined by low-frequency $Ca^{2+}$ signals are only modestly more similar to BOLD networks. In addition, the principal gradient of functional connectivity is nearly identical for BOLD and $Ca^{2+}$ signals. Despite similarities, important differences are also detected across modalities, such as in measures of functional connectivity strength and diversity. In conclusion, $Ca^{2+}$ imaging uncovers overlapping functional cortical organization in the mouse that reflects several, but not all, properties observed with fMRI-BOLD signals.

Brains show evidence of functional organization across spatio-temporal scales, from synapses to the whole organ, which varies between individuals, over time, as well as with injury or disease. Understanding the principles that govern brain organization enables their use as clinical indices. Closing knowledge gaps requires work in humans and model species, across scales, and using complementary sources of image contrast. Here, we focus on large-scale systems (i.e., networks), as a deeper understanding of their characteristics stands to have broad prognostic and diagnostic utility, in part because they can be assessed with noninvasive imaging methods that are applicable in human subjects.

Much of what we know and can access about large-scale systems, especially in humans, comes from the blood-oxygenation-level-dependent (BOLD) contrast obtained with functional magnetic resonance imaging (fMRI). Recent and growing evidence shows that measures of large-scale systems obtained with fMRI-BOLD (or proximal

optical measures of hemoglobin) are, to an extent, reflective of neural activity[1–5]. Yet, despite important progress, the relationship between fMRI-BOLD and underlying neural activity is complex and incompletely understood[6–8], which poses several challenges to interpreting network organization obtained using this technique[9–13].

A powerful tool for investigating the functional organization of large-scale networks is wide-field fluorescence imaging in mouse models bearing genetically encoded calcium ($Ca^{2+}$) sensitive indicators[14,15]. Critically, $Ca^{2+}$ imaging affords a large field of view covering much of the mouse cortical mantle and provides image contrast that is a more direct measure of neural activity than BOLD. Applied with fMRI-BOLD (or BOLD-like measures), $Ca^{2+}$ imaging can reveal the neural component captured by the BOLD signal[1,2,5,16–18]. Here, we leverage a simultaneous multimodal framework, BOLD-fMRI and $Ca^{2+}$ imaging[1], to determine both cross-modal convergent and divergent features of large-scale functional networks.

As in previous studies using similar[3,16] or the same experimental approach[1], we examine functional connectivity, a widely employed measure of inter-regional synchrony, to define and characterize large-scale brain networks. Importantly, we consider networks as having *overlapping*, rather than disjoint, functional organization. Many complex systems, including biological, technological, and social ones, are inherently overlapping (nodes participate in multiple communities or clusters) rather than *disjoint* (each node belongs to a single community)[19–21]. In the brain, overlap means that regions participate across multiple networks (to varying degrees), consistent with the notion that functionally flexible regions can contribute to multiple brain processes[22–25]. Although evidence for overlap in human brain networks has accrued based on multiple analysis techniques applied to BOLD-fMRI data[25–28], it is unclear whether the putative overlapping organization is driven, at least in part, by the nature of BOLD signals. To the best of our knowledge, the potentially overlapping functional organization of cortical networks has not been tested in animal models, where fMRI-BOLD can be obtained together with $Ca^{2+}$ signals that exhibit greater spatiotemporal resolution and capture neural activity more directly.

Here, we use highly-sampled simultaneously recorded wide-field $Ca^{2+}$ and fMRI-BOLD data to resolve whether functional networks discovered with BOLD are also detected with $Ca^{2+}$ imaging while determining their overlapping organization (Fig. 1). We use a Bayesian generative algorithm that estimates the membership strength of a given brain region to all networks[25,27,29]. Importantly, this approach also allows detection of disjoint organization in a data-driven manner. In addition, region-level properties are quantified including node degree[30,31] and diversity[32–34], while a wide range of parameters are explored to test the robustness of our findings (Table 1).

Overall, we find that overlapping network organization is robustly detected in simultaneously recorded wide-field $Ca^{2+}$ and fMRI-BOLD data regardless of the parameters selected. Evidence of rich overlapping organization advances our fundamental understanding of cortical brain organization, helping to further validate the neural origins of clinically accessible fMRI-BOLD network organization.

## Results

Mice ($n = 10$) expressing GCaMP6f in excitatory neurons underwent simultaneous wide-field $Ca^{2+}$ and BOLD-fMRI, as described previously by us (ref. 1; "Methods", Fig. 1a). Animals were lightly anesthetized (0.50–0.75% isoflurane) and head-fixed. Data were collected at each of three longitudinal sessions; each session contained four runs, each lasting 10 min for a total of 1200 min of data (Fig. 1b).

BOLD data (acquisition rate 1 Hz, "Methods") were processed using RABIES (Rodent Automated BOLD Improvement of EPI Sequences)[35–37] and high-pass filtered[38] (0.01–0.5 Hz). Given that $Ca^{2+}$ and BOLD signals are maximally correlated when $Ca^{2+}$ is temporally band-passed to match BOLD[1,2,17], and the "lowpass" nature of the BOLD

signal[39–41], we investigated network measures within a slow (BOLD-matched) and fast (0.5–5 Hz) $Ca^{2+}$ frequency range (herein, $Ca^{2+}_{slow}$ and $Ca^{2+}_{fast}$). $Ca^{2+}$ data were acquired at an effective background-corrected rate of 10 Hz and processed using a pipeline that we have published previously (ref. 42; "Methods"). Critically, we collected both GCaMP-sensitive and GCaMP-insensitive optical measurements for the removal of background fluorescence and hemoglobin signals from the $Ca^{2+}$ data (refs. 1,42–44; "Methods").

To build functional networks, a common set of regions of interest (ROIs) were defined (Fig. 1c; "Methods"). To relate 3D BOLD and 2D $Ca^{2+}$ data, we adopted the CCFv3 space for the mouse brain provided by the Allen Institute for Brain Sciences[45]. ROIs covered most of the cortex. Areas not well captured in the wide-field $Ca^{2+}$ imaging FOV were excluded (Fig. 1d). Correlation matrices were computed for each acquisition run using pairwise Pearson correlation. Matrices were binarized by retaining the top $d$% strongest edges (Table 1). We used a mixed-membership stochastic blockmodel algorithm[46] that can generate overlapping (or disjoint) networks[25,27,29]. The algorithm determines *membership* values for each ROI, with one value per network (Fig. 1e). Membership values sum to 1 across networks, which allows these values to be interpreted as probabilities. Overlapping networks, and by extension membership values, were computed at the level of runs and then averaged across sessions to determine an animal-level result. Random-effects group analysis was evaluated based on animal-level estimates and variability. Results in the main text are from 542 ROIs and $d = 15$%.

## Cortical organization captured by overlapping network solutions

Here, *network* is used interchangeably with *overlapping community*, as is *node* with *region*. Existing work has shown decomposition of the mouse cortex into as few as 2–3 networks[4,33,47], but 7–10 is more typical[36,48–50]. We explored a range of numbers of networks (3, 7, and 20). Our 3-network solution captured previously observed systems, namely, the visual (overlapping community 2, OC-2) and somatomotor (OC-3), as well as a large system (OC-1) that included territories previously classified as the mouse "default network"[49,51–53] (Fig. 2a). To facilitate comparisons to standard *disjoint* algorithms, we forced a disjoint version of our solutions by assigning each region to the network with the largest community membership value.

With 7 networks, well-defined visual and somatomotor networks (OC-2 and OC-3, respectively) were again identified[36,50], alongside additional systems covering bilateral and well-defined cortical territories (Fig. 2c). OC-1 encompassed medial areas including the cingulate cortex but also extended more laterally. OC-4 spanned from medial to lateral areas, including the somatosensory cortex. For both $Ca^{2+}_{fast}$ and $Ca^{2+}_{slow}$, OC-4 also included the frontal orienting field (FOF), a possible homolog of the frontal eye field in primates[54–58]. OC-5 largely overlapped with the anterior lateral motor area, a region involved in motor planning[43,59–61]; notably, for $Ca^{2+}_{fast}$ this network also included the supplementary somatosensory area. OC-6 overlapped with the barrel field for BOLD and $Ca^{2+}_{slow}$, but captured the upper limb somatosensory cortex for $Ca^{2+}_{fast}$. Finally, OC-7 was very different for BOLD and $Ca^{2+}$ signals; for BOLD, it was centered around FOF, and for both $Ca^{2+}$ signals it was centered around the retrosplenial cortex. We also investigated the 7-network organization in a subset of animals that underwent an additional awake imaging session that measured $Ca^{2+}$ signals outside the MRI scanner. Notably, the overall organization in awake animals (Supplementary Fig. 5) was qualitatively very similar to that obtained with lightly anesthetized animals.

The 20-network solution is shown in Supplementary Fig. 1, which revealed finer spatial networks that were again bilateral (like the 3- and 7-network solutions). Notably, even with 20 networks, the FOF did not appear as a separate network for either $Ca^{2+}$ signal, in contrast to BOLD. In sum, across solutions (3, 7, and 20 networks), recognized functional

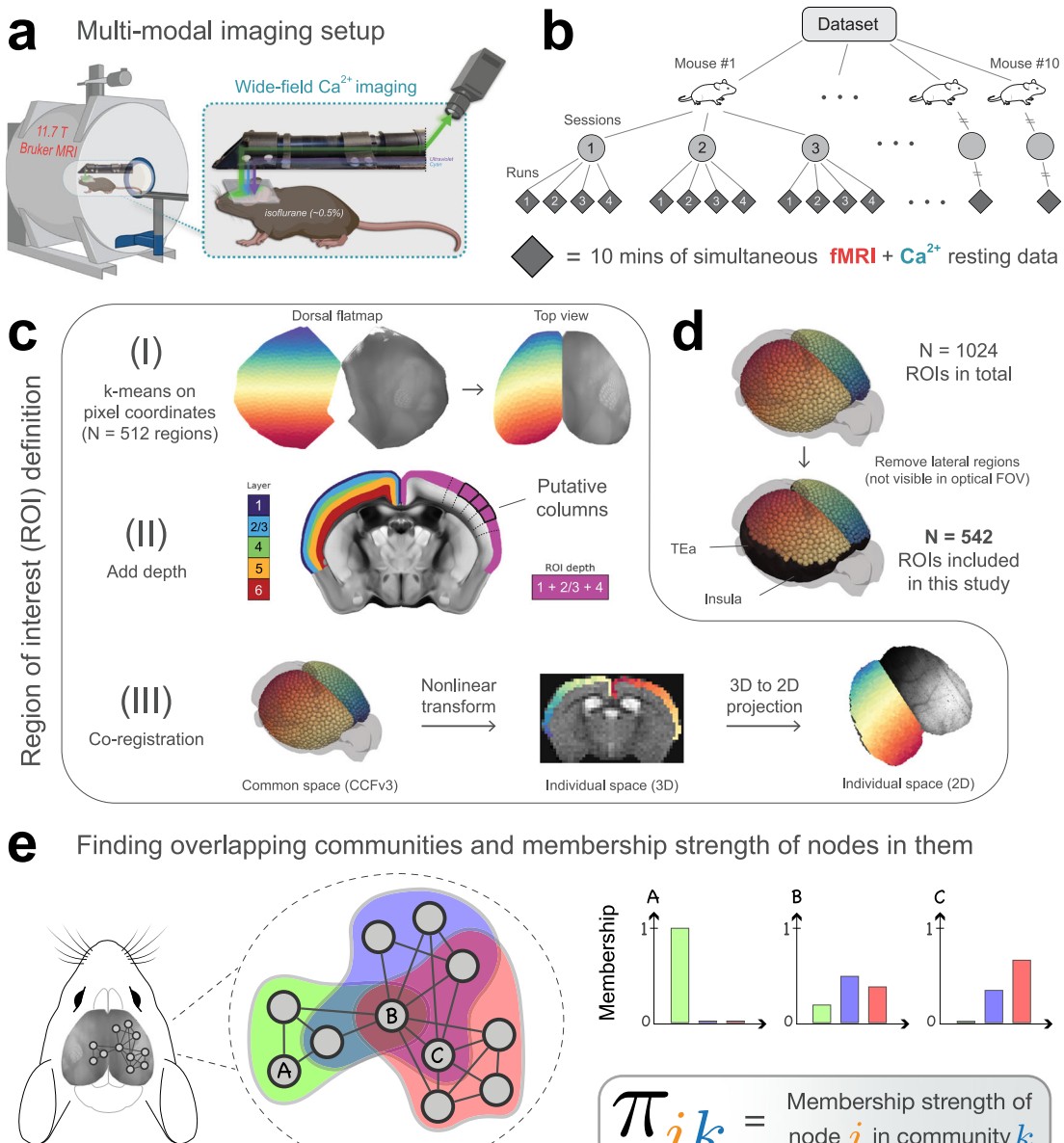

**Fig. 1 | Experimental setup and overlapping community analysis.**
**a** Simultaneous fMRI-BOLD and wide-field Ca$^{2+}$ imaging[1]. Ca$^{2+}$ data are background-corrected (illustrated by three colored wavelengths; "Methods"). **b** Hierarchical data structure. $n = 10$ mice, scanned across three longitudinal sessions, with four runs per session, each lasting 10 min. **c** Definition of ROIs within the Allen Mouse Brain Common Coordinate Framework (CCFv3)[45]. (I) Division of the mouse dorsal flatmap into $N = 1024$ spatially homogeneous ROIs. (II) Add depth by following streamlines normal to the cortical surface. The resulting ROIs are "column-like". (III) Transform ROIs from common space into 3D and 2D individual spaces ("Methods"). Dorsal flatmap, layer masks, and columnar streamlines from CCFv3.

**d** Analyses were restricted to ROIs that appeared in the Ca$^{2+}$ imaging FOV after multimodal co-registration ("Methods"). Lateral areas including the insula and temporal association areas were excluded. **e** We applied a mixed-membership stochastic blockmodel algorithm to estimate overlapping communities[29]. Membership strength (values between 0 and 1) quantifies the affiliation strength of a node in a network. Here, node A belongs only to the green community, node B belongs to all three communities with varying strengths, and node C belongs to the blue and red communities with varying strengths. **a** was created with https://BioRender.com. Mouse illustrations in (**b**) and (**e**) were downloaded from https://scidraw.io/ (refs. 122,123).

organization (established brain regions, functional networks, and a high degree of bilateral symmetry) was uncovered using our algorithm across imaging modalities.

### Intermodal network organization similarity

For 7 networks, BOLD, Ca$^{2+}_{slow}$, and Ca$^{2+}_{fast}$ were quantitatively compared (Fig. 2d–f) to test the hypothesis that band-pass filtering Ca$^{2+}$ to match BOLD leads to greater inter-modal agreement. The comparison was based on cosine similarity (1 indicates identical organization, 0.5 indicates "orthogonal/unrelated" organization, and 0 indicates perfectly "opposite" organization). The similarity between BOLD and

Ca$^{2+}_{slow}$ networks was relatively high (>0.73), except for OC-7 (0.26), a network that was evident in both Ca$^{2+}$ conditions but not captured by BOLD. In comparison, BOLD and Ca$^{2+}_{fast}$ similarity was generally lower but still relatively high for OC-1 to OC-4 (>0.77), though modest for OC-5 and OC-6 (0.59 and 0.65, respectively). Overall, band-pass filtering Ca$^{2+}$ seemed to have a modest network-dependent impact when comparing network territories across modalities.

To generate a summary metric, we collapsed across networks to generate an overall index of similarity (Fig. 2f). As expected[1–3], BOLD and Ca$^{2+}_{slow}$ solutions were more similar than BOLD and Ca$^{2+}_{fast}$ ($p < 0.05$, permutation test, Holm-Bonferroni corrected). This result

**Table 1 | Parameters explored in the present study**

| Parameter | Values and figures |
|---|---|
| Number of networks | 3 (Fig. 2a); 7 (Figs. 2–5); and 20 (Supplementary Fig. 1) |
| Initial parcellation granularity | Fine (542 regions, Fig. 1d, main paper); and Coarse (152 regions, Supplementary Fig. 2) |
| Edge density | 10–25% (Supplementary Figs. 3 and 11b) |
| fMRI preprocessing pipeline | Supplementary Fig. 11a |

To ensure the robustness of our findings, we explored a range of parameters and found that our results were qualitatively reproduced across all conditions.

was stable across 3, 7, and 20 networks, data processing parameters (Supplementary Fig. 3), including edge density, and number of ROIs (Supplementary Fig. 2).

To further probe some of the differences between BOLD and $Ca^{2+}$ results, we repeated our network similarity analysis after applying a gamma-variate hemodynamic filter[2] to the calcium signal, which we obtained from our previous work[1] (Supplementary Fig. 6). By doing so, $Ca^{2+}$ data would presumably better approximate BOLD data. Filtering $Ca^{2+}$ signals changed network organization in relatively modest ways for $Ca^{2+}_{slow}$, although some of the differences were statistically significant (Supplementary Fig. 6b). Filtering produced more pronounced quantitative changes to $Ca^{2+}_{fast}$ networks, especially OC-4, OC-5, and OC-6.

### Cortical networks show prominent overlapping organization

All results are described at the group level. However, we confirmed that the basic organization of the 7-network solution was observed at the individual level (Supplementary Fig. 4). Thus, group-level properties, including network overlap, are not driven by the process of performing group analysis.

To quantify overlapping organization, we examined the distribution of membership values across networks. Membership values range from 0–1, and sum to 1 across networks. Thus, a disjoint organization would be characterized by all regions having high membership values for a single network (a "right-peaked" distribution; Fig. 3a, left). Importantly, this outcome is observed with our algorithm when synthetic, disjoint data are simulated (see Supplementary Fig. 7). In contrast, a roughly uniform distribution of membership values would correspond to a network whose regions affiliate with multiple networks with varying strengths (Fig. 3a, middle). Finally, extreme overlap would be when regions tend to not affiliate with any network very strongly (Fig. 3a, right).

In simulations, we constructed synthetic disjoint networks with no overlap[62] and found that the algorithm detected membership values >0.8 (Supplementary Fig. 7). In other words, in a completely disjoint network, every node belongs to a single network with >0.8 strength and, given that a node's membership strength sums to 1, the remaining <0.2 strength is distributed across the remaining networks. This establishes a floor value (0.2) for robust network membership. Thus, to examine membership distributions, we considered the range (0.2, 1.0]; membership values <0.2 were not considered so as to conservatively characterize network overlap.

Across conditions (BOLD, $Ca^{2+}_{slow}$, and $Ca^{2+}_{fast}$), no more than 60% of brain regions within any network were within the "disjoint" range (>0.8). The least overlapping networks were the visual and somatomotor (OC-2 and OC-3) for all conditions and the retrosplenial network (OC-7) for the two $Ca^{2+}$ conditions. Networks with the greatest amount of overlap were OC-1 and OC-4 (Fig. 3b), which included the cingulate cortex (OC-1) as well as medial and lateral areas, including somatosensory cortex (OC-4) and the FOF (OC-4, $Ca^{2+}$).

For several of the conditions, we observed a roughly reverse L-shaped distribution (Fig. 3b). In the case of OC-2, OC-3, and OC-7 they

were reminiscent of the disjoint pattern of Fig. 3a, except that the "base" had a considerably higher level than zero. Thus, these networks have considerable disjoint organization, in some cases with 60% of the regions affiliating with a single network. OC-1, OC-5, and OC-6 exhibited a more true U shape with relatively more strongly disjoint regions (right side) and weakly affiliated regions (left side), with relatively fewer regions with intermediate membership strengths (middle two values). Nevertheless, it is important to note that in many cases the proportion of regions with intermediate membership values (0.4–0.8 range) was around 40%.

To visualize network overlap, we divided membership strength into four categories, or membership *tiers*. Because we considered 7 networks, bin thresholds were multiples of 1/7 (all statistics are FDR-corrected). Based on this representation, we observed that overlapping network organization was arranged in a spatially coherent fashion that showed a nested pattern of membership *tiers* (Fig. 4a). To quantify overlap across networks, if a region had a membership value statistically greater than 1/7 for a given network, we classified it as "belonging to" to that network. We then summed the number of networks to which regions belonged (Fig. 4b). By this definition, ~50% of brain regions belonged to more than one network (Fig. 4c). Further, brain regions belonging to more than one network were distributed across networks (Fig. 4d). In even the most disjoint-like cases (OC-2 and OC-3), >25% of regions affiliated significantly with more than one network across all conditions.

### Membership diversity reveals intermodal differences

The preceding analyses showed clear evidence for overlapping organization alongside disjoint organization in the mouse cortex across imaging modalities and frequency bands. The characteristics of this overlapping organization were further quantified using (normalized) Shannon entropy, a continuous measure of *membership diversity* computed from regional membership values ("Methods", Fig. 5a, left). A region that belongs to all networks with equal membership strengths will have maximal diversity. Conversely, a region that belongs to a single network will have minimal diversity. Thus, membership diversity is indicative of a region's multi-functionality and/or involvement in multiple processes.

The distribution of membership diversity values across all regions is shown in Fig. 5a (right). The peak near zero captures a group of regions, ~30% for $Ca^{2+}_{slow}$ and $Ca^{2+}_{fast}$, and 15% for BOLD, that are primarily associated with one network. Beyond this peak, the majority of regions displayed values more or less along a continuum, with a second smaller peak (at ~0.35) with regions affiliated with two networks (dashed line in Fig. 5a, right inset). For visualization purposes, we rank-ordered membership diversity values to inspect the overall pattern across conditions (BOLD, $Ca^{2+}_{slow}$, and $Ca^{2+}_{fast}$; Fig. 5b; for the non-rank-ordered version, see Supplementary Fig. 8a). The resulting patterns revealed modest agreement between BOLD and both $Ca^{2+}$ conditions, and especially strong agreement between the two $Ca^{2+}$ frequency bands. To quantify this agreement, we (Pearson) correlated membership diversity values: between BOLD and $Ca^{2+}_{slow}$: $r = 0.54 \pm 0.11$; BOLD and $Ca^{2+}_{fast}$: $r = 0.63 \pm 0.09$; and $Ca^{2+}_{slow}$ and $Ca^{2+}_{fast}$: $r = 0.90 \pm 0.07$ (Fig. 5c). Contrary to expectations, measures of BOLD membership diversity were not more similar to those obtained from $Ca^{2+}_{slow}$ relative to between BOLD and $Ca^{2+}_{fast}$ (Fig. 5c).

We also identified regions that showed significant differences in membership diversity magnitude between conditions by subtracting each pair of measures (Fig. 5d; FDR corrected). Spatially broad differences (BOLD versus both $Ca^{2+}$ frequency bands) were observed. Diversity was consistently larger for BOLD compared to both $Ca^{2+}$ conditions, except for two bilateral sectors that showed the opposite pattern (one in higher-order visual areas and one, for $Ca^{2+}_{slow}$ only, in a primary somatosensory area). This final observation was made alongside $Ca^{2+}_{fast}$ exhibiting a large territory of regions with higher

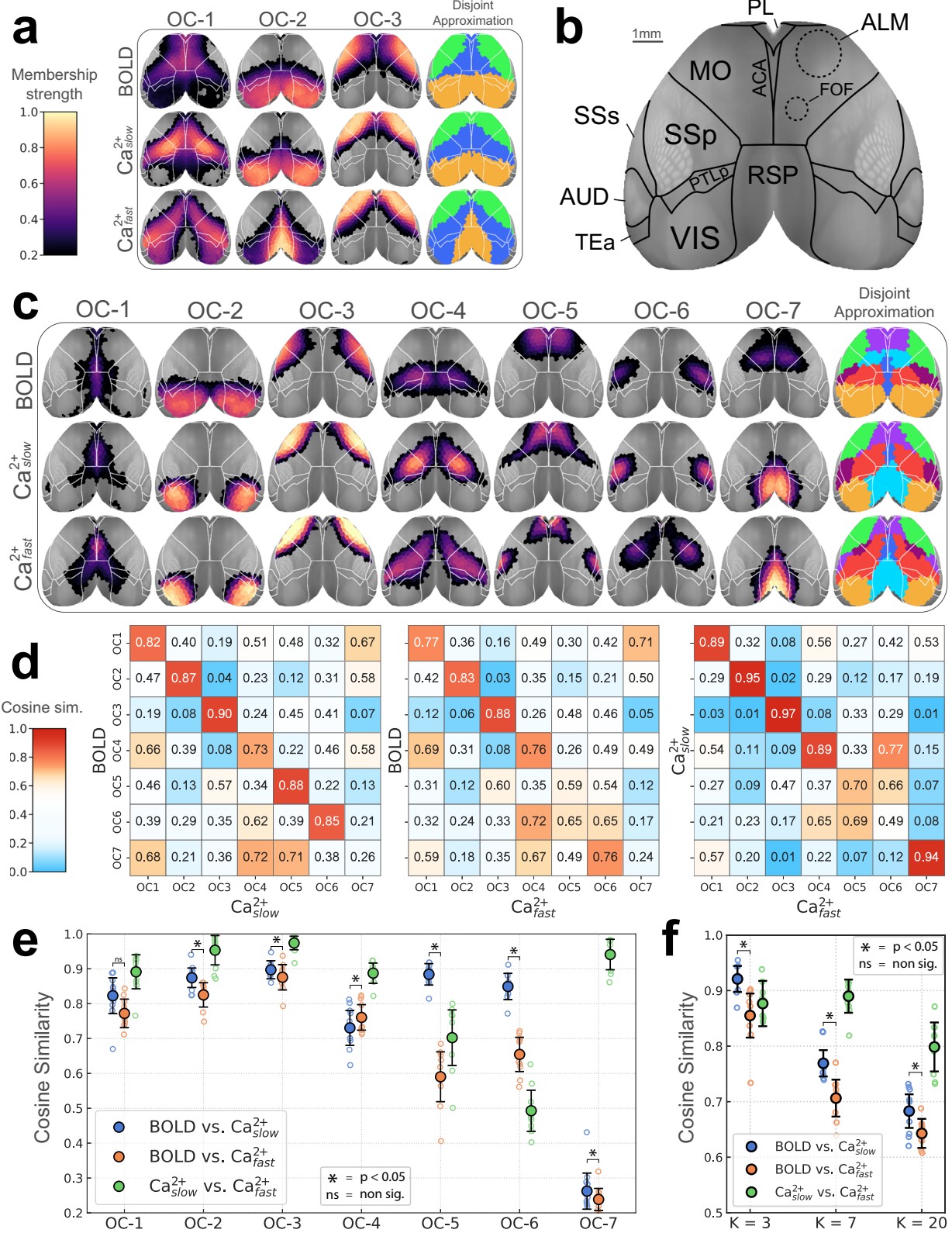

membership diversity than $Ca^{2+}_{slow}$. Overall, differences were observed despite similar proportions across modalities for values above 0.1 (Fig. 5a, right).

For a comparison between membership diversity (entropy) and participation coefficient, a measure commonly used to quantify link diversity[32–34,63,64], see Supplementary Fig. 8. We found membership diversities and participation coefficients to be in good agreement for

BOLD: $r = 0.71 \pm 0.09$, and $Ca^{2+}_{slow}$: $r = 0.77 \pm 0.12$, and to be more weakly related for $Ca^{2+}_{fast}$: $r = 0.47 \pm 0.28$.

## Region degree is substantially different across modalities

Degree is a measure of centrality that quantifies the number of functional connections of a region[30,31] (Fig. 6a, left). Importantly, degree differs from membership diversity by being independent of

**Fig. 2 | Overlapping functional networks of the mouse cortex. a** Decomposition based on 3 networks. Color scale indicates membership strengths (Fig. 1e). The disjoint approximation is obtained by taking each region's maximum membership value. **b** Cortical areas (top view) as defined in the CCFv3 Allen reference atlas[45]. In addition, dashed lines approximately correspond to functionally defined sub-regions in the secondary motor area[55,59]. **c** Decomposition with 7 networks. **d** Network similarity based on cosine similarity (1 = identical, 0.5 = "orthogonal", 0 = perfectly dissimilar or "inversely correlated"). Color scale emphasizes similarities and strong dissimilarities. **e** Diagonal elements of matrices in (**d**) are plotted. **f** Overall similarity collapsing across networks as a function of number of networks (3, 7, and 20). **e, f** Empty circles correspond to individual animals (n = 10); large solid circles are the group average. Error bars are 95% confidence intervals based on hierarchical bootstrap ("Methods"). Comparison of BOLD and $Ca^{2+}_{slow}$ networks (paired permutation test, two-sided, p < 0.05, Holm–Bonferroni corrected). The exact p values were as follows (uncorrected): OC-1, $7.4 \times 10^{-1}$, OC-2, $4.0 \times 10^{-5}$, OC-3 to 6, $2.0 \times 10^{-6}$, OC-7, $1.1 \times 10^{-2}$; and, K = 3 and 7, $2.0 \times 10^{-6}$, K = 20, $4.3 \times 10^{-3}$. OC overlapping community, ACA anterior cingulate area, ALM anterior lateral motor cortex, FOF frontal orienting field, MO somatomotor areas, PL prelimbic area, PTLp posterior parietal association areas, RSP retrosplenial area, SSp primary somatosensory area, SSs supplemental somatosensory area, VIS visual areas. See also Supplementary Figs. 1–6. Source data are provided as a Source Data file.

community (regions have functional connections both within and between communities). As in the previous section, the distribution of degree across regions was plotted for each condition (BOLD, $Ca^{2+}_{slow}$, and $Ca^{2+}_{fast}$) (Fig. 6a, right), and the spatial patterns of degree ranks were displayed on the cortex (Fig. 6b; actual values in Supplementary Fig. 9), and (Pearson) correlation was used to measure agreement between conditions (Fig. 6c).

Across conditions, degree distributions showed weak similarity (Fig. 6a, right). As expected, $Ca^{2+}_{slow}$ and $Ca^{2+}_{fast}$ were more similar to one another than to BOLD, and $Ca^{2+}_{slow}$ was more like BOLD than $Ca^{2+}_{fast}$. BOLD and both $Ca^{2+}$ measures showed differences at the low-extreme (near zero) as well as across the range: $Ca^{2+}$ having fewer low-mid degree regions, and more high-degree regions than BOLD. Intermodal differences were more pronounced when we looked at the spatial distribution of degree (Fig. 6b), and the (Pearson) correlation of degree across conditions (Fig. 6c). Similar to the case of membership diversity, degree showed a consistent spatial pattern across $Ca^{2+}$ conditions (Fig. 6b), and was highly correlated: $r = 0.87 \pm 0.08$ (Fig. 6c, right). Unlike membership diversity, BOLD and both $Ca^{2+}$ degree measures showed opposing spatial patterns (Fig. 6b), and were negatively correlated with BOLD: between BOLD and $Ca^{2+}_{slow}$: $r = -0.29 \pm 0.16$, and BOLD and $Ca^{2+}_{fast}$: $r = -0.46 \pm 0.14$ (Fig. 6c, left and middle). To account for differences based solely on the magnitude/variability of degree values, we computed percentile maps by calculating t-statistics followed by rank-ordering[33] and found the patterns to be unchanged (Supplementary Fig. 10). Further, differences across modalities persisted across edge thresholds and changes in data pre-processing steps (Supplementary Fig. 11).

### Different entropy-degree relationships across modalities
How a given brain region affiliates across multiple networks (as indexed by membership diversity/entropy) is closely linked to its roles as an integrative and/or coordination hub[34]. Furthermore, membership diversity and degree are measures that, when combined, can further uncover brain organization[32,33]. In particular, regions with low entropy and high degree have few inter-network functional connections (low entropy) and many intra-network functional connections (high degree) and can be thought of as *provincial* hubs[33,63,64]. Regions with high entropy and low degree have few functional connections but link many networks, and can be conceptualized as *connector* hubs. Inspired by the work of Yang and Leskovec[65], such organization reveals what can be called "sparse" network overlap (Fig. 7a, left). Finally, regions with high entropy and high degree interlink many networks via an organization that can be called "dense" overlap (Fig. 7a, right). To determine cortical functional organization based on these measures, we visualized region entropy-degree relationships for each condition (BOLD, $Ca^{2+}_{slow}$, and $Ca^{2+}_{fast}$) (Fig. 7b) color-coded by disjoint network assignment (Fig. 7b; inset).

Entropy and degree were inversely (Pearson) correlated for BOLD ($r = -0.44 \pm 0.16$; Fig. 7b, left). This pattern was partly driven by a concentration of regions showing sparse overlap (lower right quadrant) with connector hubs present in most networks, alongside two networks (overlapping with OC-3 and to a lesser extent OC-2; Fig. 2c),

that included regions with a more provincial hub characterization (upper left quadrant). In contrast, entropy and degree were positively correlated for $Ca^{2+}_{slow}$ ($r = 0.44 \pm 0.09$), and $Ca^{2+}_{fast}$ ($r = 0.69 \pm 0.07$) (Fig. 7b, middle and right). Like BOLD (but to a lesser extent), $Ca^{2+}_{slow}$ results included regions with sparse overlap (lower right quadrant); these overlapped with OC-6, and to a lesser extent OC-1 and OC-7 (Fig. 2c). However, unlike BOLD, there were regions with high entropy (densely overlapping regions; upper right quadrant), as well as regions with low overall functional connectivity (lower left quadrant). This pattern was more pronounced in the $Ca^{2+}_{fast}$ results where fewer regions exhibited sparse overlap (lower right quadrant; Fig. 7b, right). Together, these results uncovered distinct functional cortical organization observed with BOLD, $Ca^{2+}_{slow}$, and $Ca^{2+}_{fast}$, such that $Ca^{2+}$ signals expressed patterns of denser overlap not captured by BOLD signals.

### Nearly identical principal functional gradient across modalities
Large-scale gradients characterize brain regions along continuous axes of variation, complementing parcellation and clustering approaches that emphasize discreteness[66,67]. Previous work in mice has explored different measures of structural[68] and functional[69] gradients, and the relationships between them[47,68,69]. Here, we tested the extent to which functional connectivity gradients determined with fMRI data are replicated with $Ca^{2+}$ signals.

We estimated group-level functional connectivity matrices (separately for each modality), which were used to compute gradients ("Methods"). Here, we focus on the top four gradients (Fig. 8a), as they captured a large portion of the variance (Fig. 8b). The principal gradient (G-1) was organized in terms of primary visual cortex on one extreme and somatomotor regions on the other end, consistent with previous findings[47,69]. Notably, across conditions (BOLD, $Ca^{2+}_{slow}$, $Ca^{2+}_{fast}$), the spatial pattern of G-1 was nearly identical (Pearson r > 0.96; Fig. 8c) but the amount of variance explained (Fig. 8b) was much lower for BOLD (<5%) relative to $Ca^{2+}_{fast}$ (~17%) and especially $Ca^{2+}_{slow}$ (~30%). Although differences in spatial pattern were observed across conditions for the remaining gradients, strong similarities were also observed (note that we ordered gradients as typically done in the literature based on the magnitude of the corresponding eigenvalues); for example, BOLD G-2 was similar to G-4 for both $Ca^{2+}_{slow}$ ($r = 0.70$) and $Ca^{2+}_{fast}$ ($r = 0.77$), and BOLD G-3 was similar to $Ca^{2+}_{slow}$ G-2 ($r = 0.73$) and $Ca^{2+}_{fast}$ G-3 ($r = 0.69$). Finally, we note that BOLD G-4 exhibited a very dissimilar spatial pattern to that observed with $Ca^{2+}$ signals.

### Functional gradients of the cortex
An individual gradient can be thought of as a continuous representation of an organization feature[66], where the position of a brain region along the corresponding axis provides information about its function[67]. The overarching organization can be understood by examining how regions are situated in the space spanned by a few top gradients. We visualized this organization (Fig. 8d) as two-dimensional maps spanned by the principal gradient (G-1) and G-2/G-3 (specifically, the y coordinate refers to the G-1 value in column 1 of part a, and the x coordinate refers to the G2/G-3 value in columns 2/3 of part a).

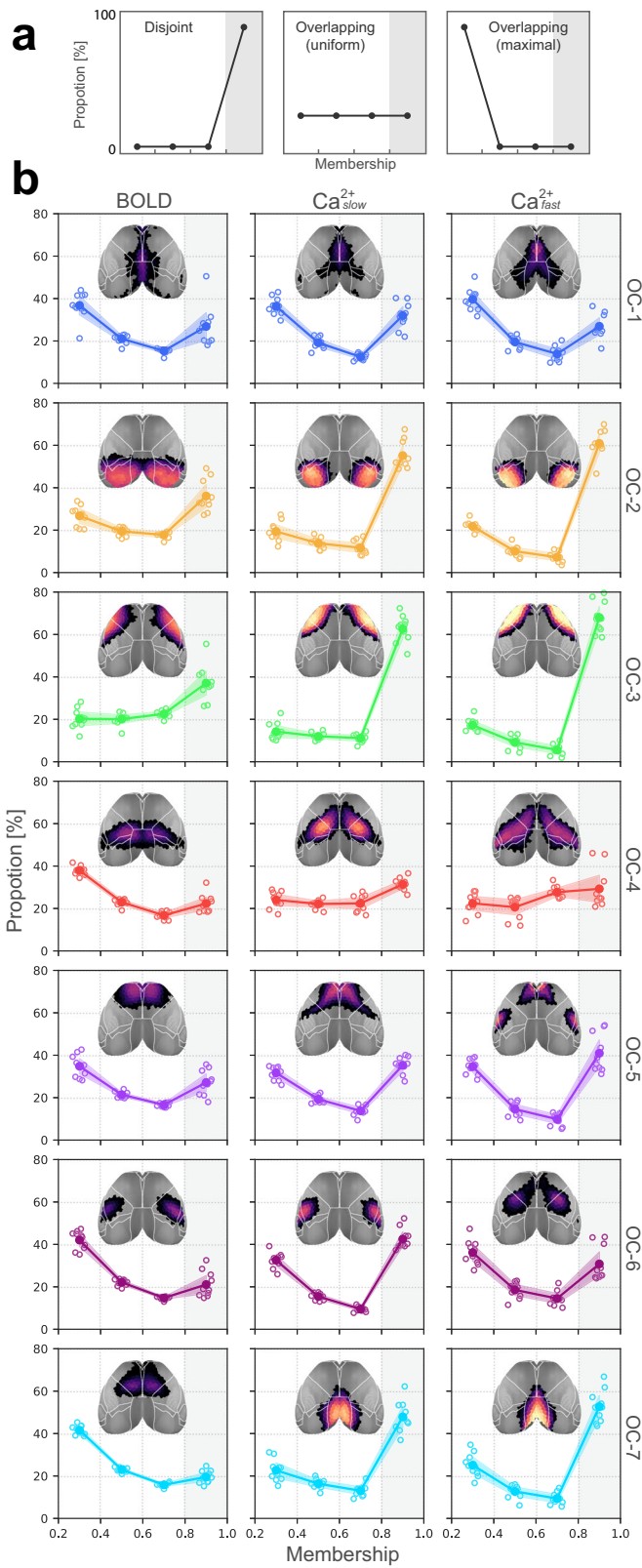

**Fig. 3 | Distribution of membership values. a** Three illustrative distributions. Left, disjoint organization; Middle, overlapping with uniform membership values; Right, completely overlapping with no mid-range or strong memberships. **b** Membership distributions computed from multimodal data indicate substantial overlapping organization. Note that the *y*-axis is capped at 80%, indicating that none of the networks are truly disjoint. Empty circles correspond to individual animals (*n* = 10); large solid circles are the group average. Error bars are 95% confidence intervals based on hierarchical bootstrap ("Methods"). OC overlapping community. Compare with Supplementary Fig. 7 for membership distributions obtained from synthetic graphs with known ground truth overlap. Source data are provided as a Source Data file.

including the anterior lateral motor cortex (purple), whereas for $Ca^{2+}$ the third anchor was the retrosplenial area (cyan). The second shape was an "arch" (panels 2, 4, and 6 of Fig. 8d) starting from visual areas on one end of the spectrum (orange), progressing toward areas usually considered part of the default network, and ending in regions within the somatomotor network (green).

## Discussion

We used recently developed simultaneous wide-field $Ca^{2+}$ and fMRI-BOLD acquisition to characterize the functional network architecture of the mouse cortex. The spatial organization of large-scale networks discovered by both modalities showed many similarities, with some temporal frequency dependence (BOLD networks were generally more similar to $Ca^{2+}_{slow}$ than $Ca^{2+}_{fast}$). Functional connectivity interrogated using a mixed-membership algorithm, instead of traditional disjoint approaches, confirmed the hypothesis that mouse cortical networks exhibit robust overlap in addition to disjoint organization when either BOLD or $Ca^{2+}$ signals were considered. Further, despite the considerable agreement, we also uncovered important differences in organizational properties across signal modalities.

Previous multimodal studies comparing cortical functional organization via concurrent GCaMP6 $Ca^{2+}$ and hemoglobin-sensitive imaging have predominantly employed seed-based analyses[3,16,70]. Such work provides information on how one or a limited set of a priori regions are functionally related to other areas but does not reveal how all regions are interrelated, which was the goal of the present work. A few studies using optical imaging have gone beyond seed-based analysis; however, the number of identified networks in these studies was limited. For example, Vanni et al.[4] investigated cortical networks in GCaMP6 mice and reported 3 networks based on slow temporal frequencies (<1 Hz) and two based on faster temporal frequencies (3 Hz) (see also ref. [5]). Here, when we decomposed the cortex into 3 networks, we observed visual (OC-2) and somatomotor (OC-3) networks and a network that overlapped with territories possibly linked to the default network (OC-1)[49,51–53] (Fig. 2a). At this coarse scale, our results agreed with Vanni et al.[4] and other seed-based approaches[3,16,36,51]. Importantly, we sought to determine functional organization at finer spatial levels, too. With 7 networks, we still observed visual (OC-2) and somatomotor (OC-3) networks, now together with a finer decomposition of other cortical systems (Fig. 2c). Overall, our analyses reproduced previous observations at a coarse scale but characterized a more fine-grained decomposition of cortical functional organization.

Next, we quantified the concordance between BOLD and $Ca^{2+}$ networks. Overall, collapsing across networks, outcomes from $Ca^{2+}_{slow}$ (BOLD-frequency matched) were more similar than $Ca^{2+}_{fast}$ to BOLD, as expected[1–3,17,39–41]. However, when networks were characterized separately, three scenarios emerged: (1) Low and high frequency $Ca^{2+}$ signals both manifested networks that were also recovered by BOLD (e.g., OC-1 to 4); (2) Low, relative to high, frequency $Ca^{2+}$ networks were a better match to their BOLD counterparts (e.g., OC-5 and OC-6); and (3) Networks that were dissimilar across modalities regardless of $Ca^{2+}$ temporal frequency (e.g., OC-7). This observation should be qualified by the finding that $Ca^{2+}_{slow}$ and $Ca^{2+}_{fast}$ results were in close agreement.

Two patterns emerged that were largely reproduced across modalities. First, a "triangular" shape with sensory and somatomotor areas on two extremes, and transmodal areas on the other extreme (panels 1, 3, and 5 of Fig. 8d). Notably, two ends of the triangular shape were anchored by visual (orange) and somatomotor (green) areas across all conditions. In contrast, the third anchoring point differed across conditions. For BOLD, it was populated by frontal areas

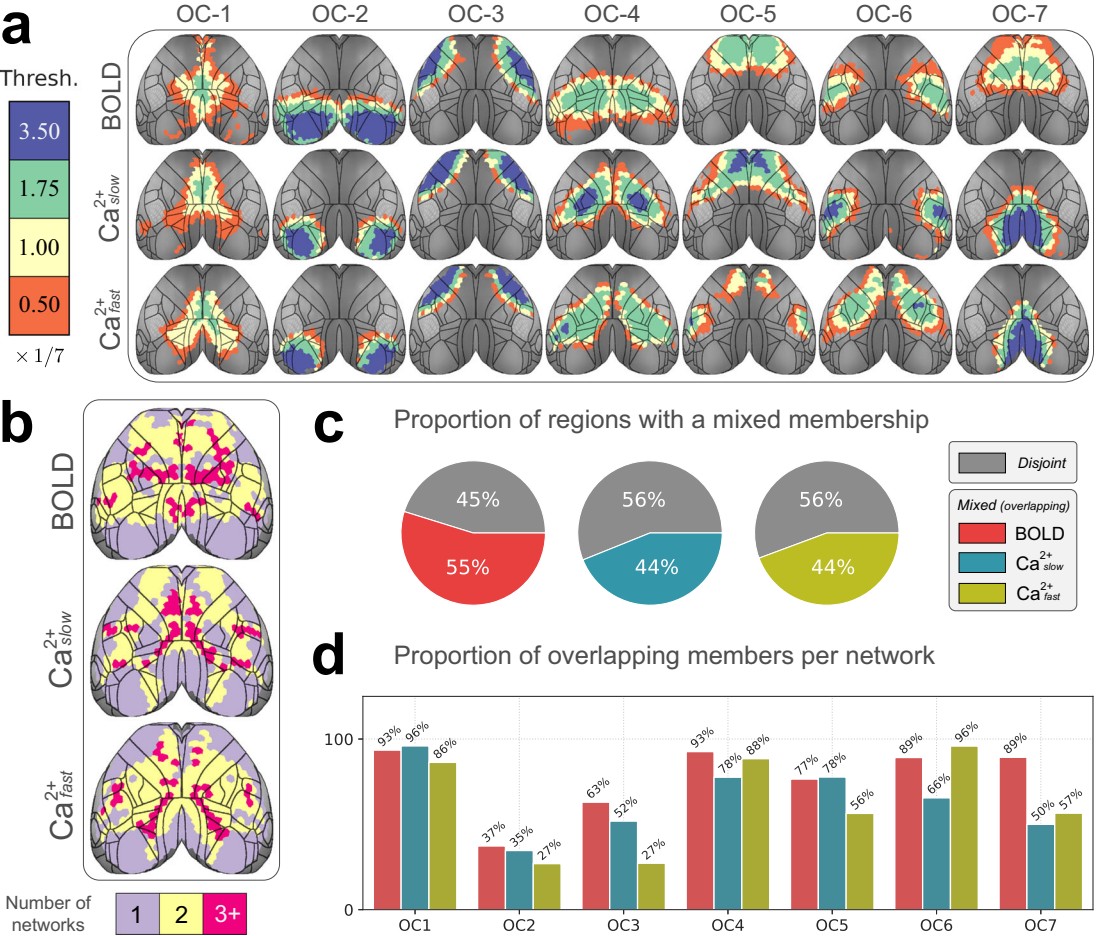

**Fig. 4 | Quantifying overlap extent. a** Membership values binned by statistical thresholding. Bins were incremented by 1/7 (for the 7 network solution). Blue (membership > 3.5 × 1/7) indicates regions with disjoint-like network affiliation. At the opposite end of the spectrum, orange (>0.05 × 1/7) indicates regions affiliated with multiple networks. Contour lines correspond to regional divisions in the Allen reference atlas (Supplementary Fig. 1b). **b** Collapsing across networks. Using a statistical threshold of 1/7 to determine if a region affiliated with a network, we counted the number of networks each region "belonged to". Most regions belonged to more than one network. **c** We defined a global overlap score as the ratio of overlapping regions divided by the total number of regions. **d** Overlap score at the network level shows that regions that affiliate with more than one network are spread across networks and present in both BOLD as well as $Ca^{2+}_{fast}$ and $Ca^{2+}_{slow}$. OC overlapping community. Source data are provided as a Source Data file.

Overall, linking the functional organization obtained with BOLD to slow $Ca^{2+}$ signals is not fully supported by our findings. In particular, the proposal that different bands capture distinct neurophysiological properties[71] was not supported for the large-scale system organization uncovered in the present work.

The pronounced differences involving network OC-7 merit particular attention (Fig. 2c). In the case of BOLD, this network was centered around parts of the secondary motor cortex known as the frontal orienting field (FOF), considered a homolog of the primate frontal eye field[54–58]. For $Ca^{2+}$ signals, FOF was consistently detected as part of a large medial network spanning somatosensory, motor, and parietal cortex (OC-4 in Fig. 2c), without forming an independent network (even in the 20-network solution; Supplementary Fig. 1). In contrast, the $Ca^{2+}$ OC-7 network was centered around the retrosplenial area in a manner that was not captured by any of the seven BOLD networks. We note, however, that the spatially finer 20-network solution for BOLD detected a retrosplenial network, although not to the same extent as identified with $Ca^{2+}$ signals (panel RSP in Supplementary Fig. 1). Overall, we speculate that the differences observed in the case of network OC-7 may reflect particularities of the signal contrasts of the two modalities. To evaluate this question, it will be useful to interrogate $Ca^{2+}$ indicators

that are sensitive to different cell populations such as inhibitory neurons (see below).

We quantified whether networks discovered for each condition (BOLD, $Ca^{2+}_{slow}$, and $Ca^{2+}_{fast}$) showed significant overlapping architecture. This was accomplished by examining the distribution of membership values and by quantifying the number of networks each region "belonged to". Notably, our algorithm detects disjoint organization in synthetic data, and the robustness of our findings was tested using a range of parameters (Table 1). Without exception, across conditions, parameter choices, and for all networks, we observed evidence of overlapping organization. On average, slightly over half of brain regions were affiliated with more than one network. Critically, although the extent of network overlap was largest for BOLD, it was also detected in $Ca^{2+}$ data regardless of temporal frequency. These results lend strong support to the validity of overlapping organization in the human brain discovered with BOLD[25–28].

Measures of network overlap consistently identified the visual (OC-2) and somatomotor (OC-3) networks as among the most disjoint across all conditions, a finding that is well aligned with their sensory and motor roles and their potential involvement in fewer brain processes. Although visual and somatomotor networks (OC-2 and OC-3)

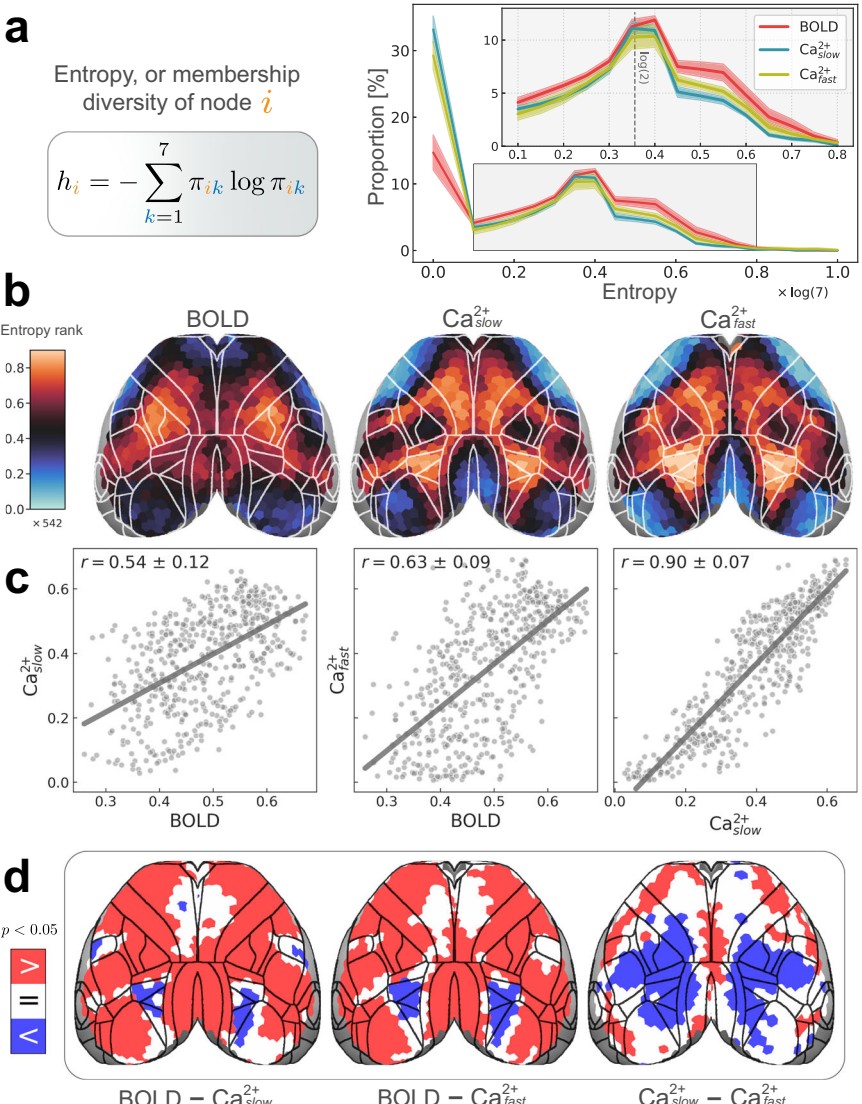

**Fig. 5 | Regional entropy, or membership diversity. a** Left: equation for Shannon entropy ("Methods"). Values are normalized [0, 1]. $h_i = 0$ if a node $i$ belongs to a single network (is disjoint); $h_i = 1$ if a node belongs to all networks with equal strength (is maximally overlapping). Right: distribution of entropies for all regions. The peak at $h = 0$ corresponds to disjoint regions. The second peak at $h \propto \log(2)$ corresponds to regions with membership values of 0.5 for two networks and 0 elsewhere. Error bars are 95% confidence intervals based on hierarchical bootstrap ("Methods"). **b** Spatial patterns of regional entropies rank-ordered (total of 542 regions) to facilitate comparisons across conditions (BOLD, $Ca^{2+}_{slow}$, and $Ca^{2+}_{fast}$). The non-rank-ordered version is shown in Supplementary Fig. 8a. Unimodal areas such as visual and somatomotor areas have low entropy (cool colors), whereas transmodal regions have high entropy (hot colors). **c** Entropy was positively (Pearson) correlated across modalities (variability obtained based on hierarchical bootstrapping; "Methods"). **d** Differences in entropy between conditions are quantified by subtracting each pair of conditions. A statistical test (paired permutation test, two-sided, $p < 0.05$, Holm–Bonferroni corrected) revealed BOLD > $Ca^{2+}$ in most regions, except for some frontal areas where BOLD = $Ca^{2+}$, and higher visual areas where BOLD < $Ca^{2+}$ (left and middle). $Ca^{2+}_{slow}$ exhibited a large territory of regions with entropy < $Ca^{2+}_{fast}$ (right). Source data are provided as a Source Data file.

exhibited the least amount of overlap among the seven networks, they were still far from being entirely disjoint. This observation is in line with previous proposals that cortical territories should be regarded as essentially multisensory[72], such that mechanisms of multisensory integration extend even into early sensory areas (see refs. [73,74]). Unexpectedly, in the case of $Ca^{2+}$ data, the retrosplenial area (OC-7) also appeared predominantly as a disjoint network. Given the retrosplenial area's recognized multisensory[75,76] and multifunctional[76,77] characteristics, further investigation is warranted to understand the underlying factors contributing to the organization detected.

Properties of network overlap, membership diversity (entropy), and degree, were quantified at the region level and compared across conditions (BOLD, $Ca^{2+}_{slow}$, $Ca^{2+}_{fast}$). As expected, $Ca^{2+}$ results showed low diversity in sensorimotor regions relative to areas that have been implicated in multiple processes and have widespread anatomical connections such as the posterior parietal cortex (PTLp; which includes higher-order visual areas)[78,79]. Despite a positive correlation with $Ca^{2+}$ results, BOLD membership diversity measures showed some peculiarities. Specifically, in contrast to $Ca^{2+}$, the posterior parietal cortex exhibited low diversity, while parts of somatosensory areas exhibited high diversity. This was unexpected given that these regions are not known to be functionally diverse (again, $Ca^{2+}$ data produced the anticipated outcome). These discrepancies did not disrupt a positive correlation between BOLD and $Ca^{2+}$ but raised questions about the extent to which the two imaging techniques are capturing the same phenomena.

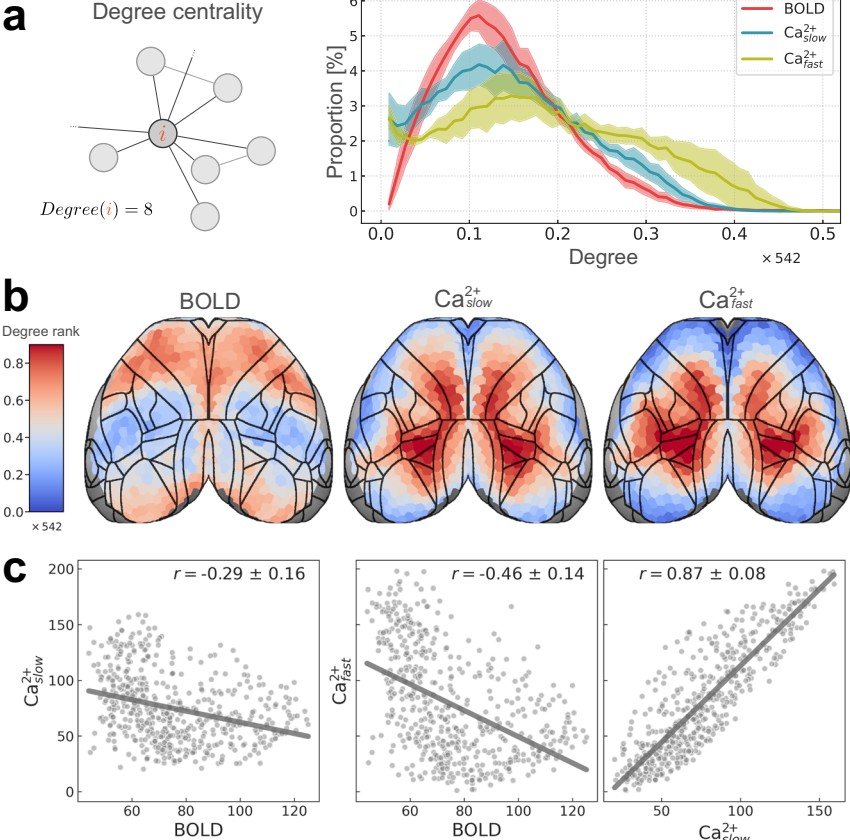

**Fig. 6 | Regional degree. a** Left: schematic of regional degree. Right: distribution of regional degree normalized by the number of brain regions (total of 542). Error bars are 95% confidence intervals based on hierarchical bootstrap ("Methods"). **b** Spatial patterns of regional degrees rank-ordered (total of 542 regions) to facilitate comparisons across conditions (BOLD, Ca²⁺slow, and Ca²⁺fast). Actual values are shown in Supplementary Fig. 9a. Densely connected regions have high degree (hot colors), whereas sparsely connected regions have low degree (cool colors). BOLD and Ca²⁺ conditions show opposing spatial patterns. **c** Similarity between conditions is quantified using (Pearson) correlation. BOLD and Ca²⁺ are negatively correlated whereas Ca²⁺ conditions are highly positively correlated (variability obtained based on hierarchical bootstrapping; "Methods"). Weighted (non-thresholded) versions of degree maps are provided in Supplementary Fig. 11. Source data are provided as a Source Data file.

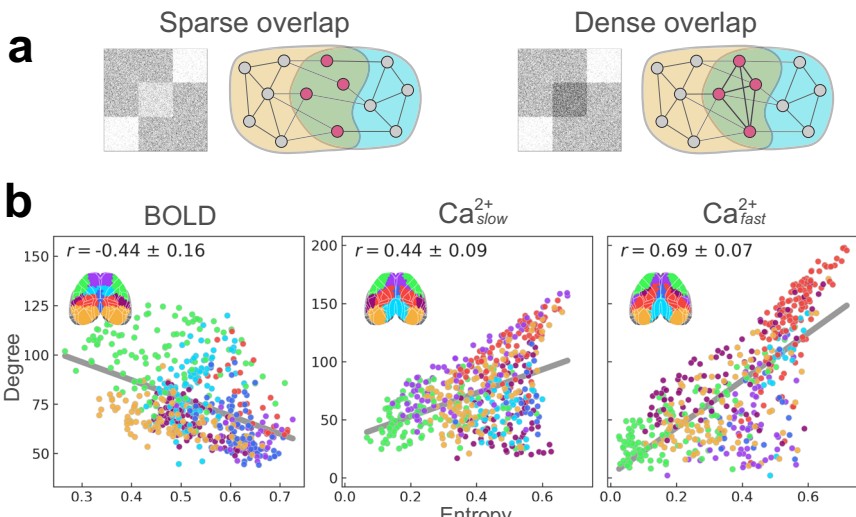

**Fig. 7 | Entropy-degree relationships across modalities. a** Illustrated examples of sparse and dense overlapping organization. Figure inspired by Yang and Leskovec[65]. **b** Entropy versus degree for BOLD (left), Ca²⁺slow (middle), and Ca²⁺fast (right). Each point corresponds to a brain region, color-coded by their disjoint network assignment (inset; see last column in Fig. 2c). See also Supplementary Fig. 8. Source data are provided as a Source Data file.

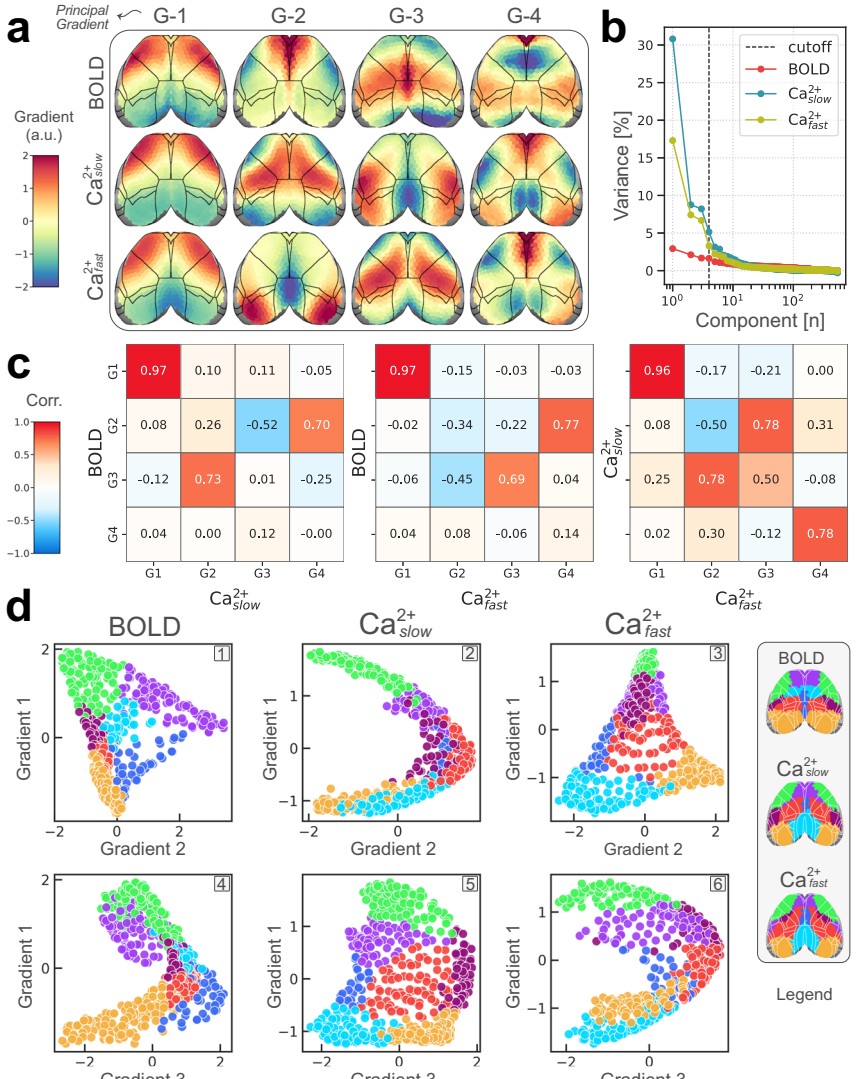

**Fig. 8 | Functional connectivity gradients. a** Top four gradients are visualized (*z*-scored; see "Methods"), and ordered based on the magnitude of the corresponding eigenvalues[109]. **b** Portion of variance explained. **c** Pearson correlations between all pairs of gradients shown in part (**a**). **d** Scatter plots display relationships between pairs of gradient axes: the *y* coordinate refers to the G-1 value in column 1 in (**a**), and the *x* coordinate refers to the G2/G-3 value in columns 2/3 in (**a**). Similar to Fig. 7b,

each point is a brain region color-coded according to its disjoint network assignments (legend from last column of Fig. 2c). Two patterns emerged. A "triangular" shape (panels 1, 3, 5), reminiscent of previous work in mice[47] and humans[67], and an "arch" (panels 2, 4, 6) connecting sensory and somatomotor areas on two ends through transmodal areas in the middle. Source data are provided as a Source Data file.

The story became more complicated when we examined region degree, a measure of centrality used at times as an indicator of region "hubness" and/or importance. Degree showed a negative correlation between BOLD and Ca²⁺ measurements with clearly different spatial patterns. Ca²⁺ degree maps highlighted association areas such as the posterior parietal cortex (PTLp), an area that is anatomically and functionally thought to be important for multimodal integration[78]. In contrast, BOLD degree maps produced counter-intuitive results where, for example, lateral somatomotor areas exhibited high degrees though they are likely to be involved in relatively fewer processes. Thus, it appears that while Ca²⁺ degree maps capture region "importance" well, the same cannot be said for BOLD. Indeed, the present finding resonates with previous suggestions that degree centrality alone should not be used to determine the hubness status of brain areas[32], particularly in the case of BOLD data.

The relationship between entropy and degree helped to uncover additional properties of cortical functional organization. For BOLD, membership diversity and degree were inversely correlated, a pattern indicative of "sparse overlap" alongside some networks that included

"provincial" hubs. Notably, this pattern has been observed in human BOLD data[32,34]. In contrast, Ca²⁺ data exhibited a positive correlation (that was more pronounced for Ca²⁺_fast), suggestive of "dense overlap" alongside regions showing few connections. Together, these results indicated that BOLD and Ca²⁺ capture distinct forms of overlapping network organization, with Ca²⁺ signals particularly able to uncover a "diverse club" of regions[34] that are densely overlapping[65], reminiscent of the "communication core" organization of structural connections in nonhuman primates[80,81].

Functional gradients characterize continuous axes of variation and/or organization across the cortex[66]. Here, we investigated the top four gradients, which revealed that the principal gradient was nearly identical across modalities, and captured a visual-to-somatomotor axis (Fig. 8). Indeed, other studies have identified this gradient as a key organizational feature of the mouse cortex[47,69]. Two other gradients (G-2 and G-3) were also well captured across modalities, although their ordering (based on eigenvalues) was different (but note that the percentage of variance explained was very similar). Finally, the spatial organization of BOLD G-4 was not reflected in the gradients identified

based on $Ca^{2+}$ signals. Intriguingly, this gradient highlights regions overlapping with the frontal orienting field captured by BOLD OC-7 (Fig. 2c), a network that was not detected by $Ca^{2+}$ even in the case of a 20-network decomposition (Supplementary Fig. 1). It is also noteworthy that some of the gradients detected spatial organization that was not reflected in the network organization maps (Fig. 2). Overall, our results unveiled three functional gradients that exhibited strong to intermediate correspondence across conditions, as well as one spatial organization seen in BOLD that was not present in $Ca^{2+}$ signals.

It is possible that some of the discrepancies between BOLD and $Ca^{2+}$ results stemmed from $Ca^{2+}$ signals originating from excitatory neurons, while the BOLD signal is cell-type agnostic. Despite excitatory neurons being the most populous cell type in the cortex, it is still unclear to what extent the activity of other cell populations such as inhibitory neurons[82,83] and astrocytes[84], or vascular effects[7,8,85,86], influence the BOLD signal. This will be explored in our future work utilizing the methods established here. Another important consideration is our use of anesthesia. Due to the challenges of imaging awake mice[87], especially head motion, we opted to use low levels of anesthesia. Head motion systematically alters the correlation structure of functional data and was therefore of particular concern in our analyses[88,89] ("Methods"). However, the effects of anesthesia on brain activity and neurovascular coupling are complex and may vary by region, anesthetic agent, and dose[70,90–92]. Our future studies will evaluate how functional networks, and their properties, differ between awake and anesthetized animals. Notably, the effects of brain state on functional organization were minimized because the same "ground truth" brain activity underlies the results from each modality, given the simultaneous nature of our multimodal data. Thus, moment-to-moment brain-state differences were not driving factors behind our findings. Nevertheless, to evaluate the impact of anesthesia on our results, we performed initial, exploratory analyses in a group of $N = 5$ animals for which we had $Ca^{2+}$ recordings in both anesthetized and awake states. These animals, which participated in the sessions reported in the main results, underwent an additional $Ca^{2+}$ recording session outside of the scanner while awake. The results (Supplementary Fig. 5) clearly show that the large-scale network organization that we reported in the lightly anesthetized state is also observed in awake mice. Relatively minor changes were also observed but require a larger sample for proper statistical comparisons. Nevertheless, the findings with this small group of animals considerably strengthen the generalizability of our findings.

We note that the analysis methods we used do not strictly require simultaneous data collection. Further, although the findings reported in the main text were at the group level, our highly-sampled dataset allowed network organization to be determined at the level of the individual, lending considerable strength to our group-level findings, and underscoring the translational potential of our approach[93]. Future work will further explore individualized network properties while exploiting the simultaneous nature of these data.

Processing and analyzing multimodal data entails making several parameter choices that potentially affect outcome measures. In particular, network overlap could be inflated by spatial misalignment. We took great care in co-registering our data and optimizing our parameter set (from the Advanced Normalization Tools package[94]). Further, issues of misalignment were considerably reduced by estimating network measures at the level of runs and combining values subsequently. Thus, modest misalignment after registration did not inflate the overall evaluation of overlap ("Methods"). In addition, the quantification of membership strength was applied to values that were thresholded based on statistical significance. We also used relatively sparse graphs (15% density in the main text), such that only the strongest correlations were considered; further analyses that quantified the extent of overlap considered only membership values that statistically exceeded 1/7 (for the 7-network solution). We also probed

the effects of parameter changes (Table 1) and found that our results were qualitatively robust. Finally, we emphasize that, while we believe our results provide evidence for considerable network overlap, estimating the extent of overlap quantitatively remains a challenging problem in network science. Important progress in this direction involves developing principled approaches based on statistical inference and generative modeling[95,96].

In conclusion, we employed simultaneous wide-field $Ca^{2+}$ imaging and fMRI-BOLD in a highly sampled group of mice expressing GCaMP6f in excitatory neurons to determine the relationship between large-scale networks discovered by the two techniques. Our findings demonstrated that (1) most BOLD networks were detected via $Ca^{2+}$ signals. (2) Considerable overlapping, in addition to disjoint, network organization was recovered from both modalities. (3) The large-scale functional organization determined by $Ca^{2+}$ signals at low temporal frequencies (0.01–0.5 Hz), relative to high frequencies (0.5–5 Hz), was more similar to those recovered with BOLD. (4) The principal functional connectivity gradient was nearly identical across all modalities, yet, quantitative and qualitative differences were also observed across gradients. (5) Key differences were uncovered between the two modalities in the spatial distribution of membership diversity and the relationship between region entropy (i.e., network affiliation diversity) and degree. Together these findings uncovered a distinct overlapping network phenotype across modalities. In sum, this work revealed that the mouse cortex is functionally organized in terms of overlapping large-scale networks that are observed with BOLD, lending fundamental support for the neural basis of such a property, which is also observed in human subjects. The robust differences that were uncovered demonstrate that $Ca^{2+}$ and BOLD also capture some complementary features of brain organization. Future work exploring these commonalities and differences, using the simultaneous multimodal acquisition used here, promises to help uncover how large-scale networks are supported by underlying brain signals in health and disease.

## Methods

### Experimental model and subject details
All procedures were approved by the Yale Institutional Animal Care and Use Committee (IACUC) and followed the National Institute of Health Guide for the Care and Use of Laboratory Animals. All surgeries were performed under anesthesia.

**Animals**. Mice ($n = 10$) were group-housed on a 12-h light/dark cycle with ad libitum food and water. Cages were individually ventilated. As per IACUC policy, at all locations where mice were housed, the temperature was between 68–79°F and humidity was between 30–70%. Animals were 6–8 weeks old, 25–30 g, at the time of the first imaging session. We explicitly conducted our study on a mixed-sex sample but did not consider sex as an independent variable given the small sample size. Although both male and female mice were used, sex information was not recorded. Animals (*SLC*, *Slc17a7-cre/Camk2α-tTA/TITL-GCaMP6f* also known as *Slc17a7-cre/Camk2α-tTA/Ai93*) were generated from parent 1 (*Slc17a7-IRES2-Cre-D*) and parent 2 (*Ai93(TITL-GCaMP6f)-D;CaMK2a-tTA*). Both were on a C57BL/6J background. To generate these animals, male CRE mice were selected from the offspring of parents with different genotypes, which is necessary to avoid leaking of CRE expression. Animals were originally purchased from the Jackson Laboratory.

**Head-plate surgery**. All mice underwent a minimally invasive surgical procedure enabling permanent optical access to the cortical surface[1]. Mice were anesthetized with 5% isoflurane (70/30 medical air/$O_2$) and head-fixed in a stereotaxic frame (KOPF). After immobilization, isoflurane was reduced to 2%. Paralube was applied to the eyes to prevent dryness, meloxicam (2 mg/kg body weight) was administered

subcutaneously, and bupivacaine (0.1%) was injected under the scalp (incision site). Hair was removed (NairTM) from the surgical site and the scalp was washed with betadine followed by ethanol 70% (three times). The scalp was removed along with the soft tissue overlying the skull and the upper portion of the neck muscle. Exposed skull tissue was cleaned and dried. Antibiotic powder (Neo-Predef) was applied to the incision site, and isoflurane was further reduced to 1.5%. Skull-thinning of the frontal and parietal skull plates was performed using a hand-held drill (FST, tip diameter: 1.4 and 0.7 mm). Superglue (Loctite) was applied to the exposed skull, followed by transparent dental cement (C&B Metabond®, Parkell) once the glue dried. A custom in-house-built head plate was affixed using dental cement. The head-plate was composed of a double-dovetail plastic frame (acrylonitrile buta-diene styrene plastic, TAZ-5 printer, 0.35 mm nozzle, Lulzbot) and a hand-cut microscope slide designed to match the size and shape of the mouse skull. Mice were allotted at least 7 days to recover from head-plate implant surgery before undergoing the first of three multimodal imaging sessions.

## Multimodal image acquisition

All mice, $n = 10$, underwent 3 multimodal imaging sessions with a minimum of 7 days between acquisitions. All animals underwent all imaging sessions. None were excluded prior to the study end-point. Data exclusion (based on motion etc.) is described below. During each acquisition, we simultaneously acquired fMRI-BOLD and wide-field $Ca^{2+}$ imaging data using a custom apparatus and imaging protocol[1]. Functional MRI data were acquired on an $11.7T$ system (Bruker, Bill-erica, MA), using ParaVision version 6.0.1 software. During each imaging session, 4 functional resting-state runs (10 min each) were acquired. In addition, 3 runs (10 min each) of unilateral light stimulation data were acquired. These data are not used in the present study. Structural MRI data were acquired to allow both multimodal registration and registration to a common space. Mice were scanned while lightly anesthetized (0.5–0.75% isoflurane in 30/70 $O_2$/medical air) and freely breathing. Body temperature was monitored (Neoptix fiber) and maintained with a circulating water bath.

**Functional MRI.** We employed a gradient-echo, echo-planar-imaging sequence with a 1.0 s repetition time (TR) and 9.1 ms echo time (TE). Isotropic data (0.4 mm × 0.4 mm × 0.4 mm) were acquired along 28 slices providing near whole-brain coverage.

**Structural MRI.** We acquired four structural images for multimodal data registration and registration to a common space. (1) A multi-spin-multi-echo (MSME) image sharing the same FOV as the fMRI data, with a TR/TE of 2500/20 ms, 28 slices, two averages, and a resolution of 0.1 mm × 0.1 mm × 0.4 mm. (2) A whole-brain isotropic (0.2 mm × 0.2 mm × 0.2 mm) 3D MSME image with a TR/TE of 5500/20 ms, 78 slices, and two averages. (3) A fast-low-angle-shot (FLASH) time-of-flight (TOF) angiogram with a TR/TE of 130/4 ms, resolution of 0.05 mm × 0.05 mm × 0.05 mm and FOV of 2.0 cm × 1.0 cm × 2.5 cm (positioned to capture the cortical surface). (4) A FLASH image of the angiogram FOV, including four averages, with a TR/TE of 61/7 ms, and resolution of 0.1 mm × 0.1 mm × 0.1 mm.

**Wide-field fluorescence $Ca^{2+}$ imaging.** Data were recorded using CamWare version 3.17 at an effective rate of 10 Hz. To enable frame-by-frame background correction, cyan (470/24, $Ca^{2+}$-sensitive) and violet (395/25, $Ca^{2+}$-insensitive) illumination (controlled by an LLE 7Ch Controller from Lumencor) were interleaved at a rate of 20 Hz. The exposure time for each channel (violet and cyan) was 40 ms to avoid artifacts caused by the rolling shutter refreshing. Thus, the sequence was: 10 ms blank, 40 ms violet, 10 ms blank, 40 ms cyan, and so on. The custom-built optical components used for in-scanner wide-field $Ca^{2+}$ imaging have been described previously[1].

## Image preprocessing

**Multimodal data registration.** All steps were executed using tools in BioImage Suite (BIS) specifically designed for this purpose[1]. For each animal, and each imaging session, the MR angiogram was masked and used to generate a view that recapitulates what the cortical surface would look like in 2D from above. This treatment of the data highlights the vascular architecture on the surface of the brain (notably the projections of the middle cerebral arteries, MCA) which are also visible in the static wide-field $Ca^{2+}$ imaging data. Using these and other anatomical landmarks, we generated a linear transform that aligns the MR and wide-field $Ca^{2+}$ imaging data. The same static wide-field image was used as a reference for correcting motion in the time series. To register the anatomical and functional MRI data, linear transforms were generated and then concatenated before being applied.

Data were registered to a reference space (CCFv3[45]) using iso-tropic whole-brain MSME images via affine followed by non-linear registration (ANTS, Advanced normalization tools[94]). The histological volume in CCFv3 was used because of a better contrast match with MRI images. The goodness of fit was quantified using mutual information and a hemispheric symmetry score that captured the bilateral symmetry of major brain structures. A large combination of registration hyperparameters was explored, and the top 10 fits per animal were selected. The best transformation out of this pool was selected for each animal by visual inspection.

**RABIES fMRI data preprocessing.** For fMRI preprocessing, we used RABIES (Rodent automated BOLD improvement of EPI sequences) v0.4.2[35]. We applied functional inhomogeneity correction N3 (non-parametric nonuniform intensity normalization[97,98]), motion correction (ANTS, Advanced normalization tools[94]), and slice time correction, all in native space. A within-dataset common space was created by nonlinearly registering and averaging the isotropic MSME anatomical images (one from each mouse at each session), which was registered to the Allen CCFv3 reference space using a nonlinear transformation (see above).

For each run, fMRI data were motion-corrected and averaged to create a representative mean image. Each frame in the time series was registered to this reference. To move the fMRI data to the common space, the representative mean image was registered to the isotropic structural MSME image acquired during the same imaging session. This procedure minimizes the effects of distortions caused by susceptibility artifacts[99]. Then, the three transforms—(1) representative mean to individual mouse/session isotropic MSME image, (2) individual mouse/session isotropic MSME image to within-dataset common space, and (3) within-dataset common space to out-of-sample common space—were concatenated and applied to the fMRI data. Functional data (0.4 mm isotropic) were upsampled to match anatomical MR image resolution (0.2 mm isotropic). Registration performance was visually inspected and verified for all sessions. Motion was regressed (6 parameters). Current best practices in both human[38] and mouse[35] fMRI preprocessing include the application of a high-pass or band-pass filter to remove physiological and other sources of noise. Specifically, high-pass filtering removes slow (<0.01 Hz) drifts from data[38]. Therefore, a high-pass filter (3rd order Butterworth) between [0.01–0.5]Hz was applied, and 15 time points (15 s of data) were discarded from both the beginning and the end of the time series to avoid filtering-related edge artifacts. Finally, average white matter and ventricle time courses were regressed.

**Fluorescence $Ca^{2+}$ imaging data preprocessing.** The raw signal was split between GCaMP-sensitive and GCaMP-insensitive imaging frames. Spatial smoothing with a large kernel (16-pixel kernel, median filter) was applied to reduce and/or remove focal artifacts (e.g., dust or dead pixels from broken fibers). Focal artifacts do not move with the subject and can bias motion correction. Motion correction parameters

were estimated on these data using normalized mutual information. Rigid image registration was performed between each imaging frame in the time series and the reference frame. Registration parameters were saved, and the large kernel-smoothed images were discarded. Modest spatial smoothing (4-pixel kernel, median filter) was applied to the raw data, and these data were motion-corrected by applying the parameters estimated in the previous step. Data were down-sampled by a factor of two in both spatial dimensions, which yielded a per pixel spatial resolution of $50 \times 50\,\mu m^2$ (original was $25 \times 25\,\mu m^2$). Photo bleach correction was applied to reduce the exponential decay in fluorescence at the onset of imaging[100]. The fluorophore-insensitive time series were regressed from the fluorophore-sensitive time series. The first 50 s of data were discarded due to the persistent effects of photobleaching in the $Ca^{2+}$ data[42]. fMRI-BOLD and wide-field calcium imaging have unmatched temporal sampling rates (1 versus 10 Hz) and different sources of noise. Accordingly, a band-pass filter (3rd order Butterworth) was applied that matched the frequencies of BOLD and $Ca^{2+}_{slow}$ ([0.01,0.5] Hz), and a complementary band with higher frequencies ($Ca^{2+}_{fast}$: [0.5,5.0] Hz). The application of band-passing also aids with the removal of noise, including slow drifts. Finally, 15 time points (1.5 s of data) were discarded from both the beginning and the end of the time series to avoid filtering-related edge artifacts[89].

**Frame censoring.** Data were scrubbed for motion using a conservative 0.1 mm threshold. High-motion frames were selected based on estimates from the fMRI time series and applied to both fMRI and $Ca^{2+}$ data. Runs were removed from the data pool if half of the imaging frames exceeded this threshold for a given run. In this dataset, 2 runs (or ~1.7% of all runs) were removed for this reason. Additionally, two more runs were removed because they did not pass our quality control criteria.

## Parcellating the cortex into columnar regions of interest (ROI)

To create regions of interest (ROIs), we employed the Allen CCFv3 (2017) reference space[45] and used their anatomical delineations as our initial choice of ROIs. However, this led to poor performance (see Supplementary Discussion). Here, we introduce a spatially homogeneous parcellation of the mouse cortex that can be adopted for both 3D fMRI and 2D wide-field $Ca^{2+}$ imaging data.

The procedure worked as follows. (1) We generated a cortical flatmap within the CCFv3 space using code published by Knox et al.[101] (link provided in "Code availability" section). (2) We subdivided the left hemisphere into $N$ regions via $k$-means clustering applied to pixel coordinates (for most analyses reported, $N = 512$). The right hemisphere was obtained by simple mirror-reversal to obtain a total of $2N$ regions. (3) Depth was added to the ROIs to obtain column-shaped regions. To do so, a path was generated by following streamlines normal to the surface descending in the direction of white matter (streamline paths were available at 10 μm resolution in CCFv3; see Fig. 3F in ref. 45). Here, we chose ROI depths so that we included potential signals from approximate layers 1 to 4 (layer masks were obtained from CCFv3). Evidence from wide-field $Ca^{2+}$ imaging suggests that signals originate from superficial layers but can extend into the cortex to some extent[15,43,102–104]. (4) Finally, ROIs were downsampled from 10 μm to 100 μm resolution, see Fig. 1c.

After co-registration, ROIs were transformed from the CCFv3 space into each individual's 3D and 2D anatomical spaces (see above). On average, ROIs had a size of $8 \pm 3$ voxels (3D, fMRI) and $48 \pm 20$ pixels (2D, $Ca^{2+}$) in individual spaces (mean ± standard deviation).

## Functional network construction

Time series data were extracted and averaged from all voxels/pixels within an ROI in native space to generate a representative time series per ROI. For each modality, for each run, an adjacency matrix was calculated by applying Pearson correlation to time series data to each ROI pair. Next, we binarized the adjacency matrices by rank ordering the connection weights and maintaining the top 15%; thus, after binarization, the resulting graphs had a fixed density of $d = 15\%$ across runs and modalities. This approach aims to keep the density of links fixed across individuals and runs and better preserves network properties compared to absolute thresholding[105]. To establish the robustness of our results to threshold values, we also tested values of 10% to 25% in 5% increments.

## Finding overlapping communities

Overlapping network analysis was applied by using SVINET, a mixed-membership stochastic blockmodel algorithm[29,106], which has been previously applied to human fMRI data by us[25] and other groups[27]. SVINET models the observed graph within a latent variable framework by assuming that the existence (or non-existence) of links between pairs of nodes can be explained by their latent community memberships. For binary adjacency matrix $A$ and membership matrix $\pi$, the model assumes the conditional probability of a connection as follows:

$$p(A_{ij}|\pi_i,\pi_j) \propto \sum_{k=1}^{K} \pi_{ik}\pi_{jk}, \qquad (1)$$

where $K$ is the number of communities, and $A_{ij} = 1$ if nodes $i$ and $j$ are connected and 0 otherwise. Intuitively, pairs of nodes are more likely to be connected if they belong to the same community or to (possibly several) overlapping communities. More formally, SVINET assumes the following generative process:

(1)  For each node, draw community memberships $\pi_i \sim$ Dirichlet($\alpha$)
(2)  For each pair of nodes $i$ and $j$:
- Draw community indicator $z_{i \to j} \sim \pi_i$
- Draw community indicator $z_{i \leftarrow j} \sim \pi_j$
- Assign link between $i$ and $j$ if $z_{i \to j} = z_{i \leftarrow j}$.

Model parameters $\alpha$ are fit using stochastic gradient ascent[107,108]. The algorithm was applied to data from each run using 500 different random seeds. Results across seeds were combined to obtain a final consensus for a run.

**Aligning community results.** Communities were identified in random order due to the stochastic nature of our algorithm. Maximum cosine similarity of the cluster centroids was used to match communities across calculations (runs or random seeds), as follows. For each run, membership vectors from all random seeds were submitted to $k$-means clustering (sklearn.cluster.KMeans) to determine $K$ clusters (e.g., $K = 7$ for analyses with 7 communities). The similarity between membership vectors from each random seed (source) and the cluster centroids (target) was then established via pairwise cosine similarity, yielding a $K \times K$ similarity matrix from source to target per random seed. A permutation of the rows of this similarity matrix was identified such that diagonals had maximum average similarity, a procedure known as the Hungarian algorithm. Finally, the identified permutation was applied to seed results, thereby aligning them with the targets. The aligned communities were then averaged. The outcome was membership matrix $\pi$ (Fig. 1e). Importantly, this matching procedure was done for each condition (BOLD, $Ca^{2+}_{slow}$, $Ca^{2+}_{fast}$) separately.

## Group results

Crucially, all measures were computed at the run level first before combining at the group level.

**Membership matrices.** This is what's visualized in Fig. 2a, c and Supplementary Figs. 1 and 2a, c.

**Thresholding membership values.** To enhance the robustness of our estimates of network overlap, membership values were thresholded to

zero if they did not pass a test rejecting the null hypothesis that the value was zero. After thresholding, the surviving membership values were rescaled to sum to 1. Thresholding was performed for each animal separately by performing a one-sample *t*-test and employing a false discovery rate of 5%. All results shown utilized this step, with the exception of figures that illustrate the spatial patterns of membership values (and do not estimate network overlap). Note that almost all (~99%) memberships that did not reach significance had values in the range [0, 0.2].

**Region functional diversity.** Shannon entropy was applied to membership matrices for each run separately before averaging. That is, given a membership matrix $\pi$ from a run, node entropies were computed to get an entropy estimate per node at the run level:

$$\text{(normalized) entropy of node } i \text{ at run level} := h_i = f(\pi_i) = -\sum_{k=1}^{K} \pi_{ik} \log_K \pi_{ik}, \tag{2}$$

where $K = 7$ is the number of communities and $\log_K$ is logarithm in base $K$. Entropy values were combined by averaging across runs to get the group-level estimates. This is what's visualized in Fig. 5b. Similarly, group averages were used to calculate the correlations between modalities in Fig. 5c.

**Computing distributions.** Similar to the above, distributions were computed for each run separately before combining at the group level. For example, consider $h_i$ (entropy of node $i$) from a run. We computed percentage values using 20 bins of width 0.05 that covered the entire range of normalized entropy values [0, 1]. We then averaged over run-level histogram values to get group-level estimates shown in Fig. 5a. Other distributions were computed in an identical way. Specifically, 57 bins of size 5 were used for Fig. 6a, and 4 bins of size 0.2 were used for Fig. 3b.

**Gradient analysis**
To obtain functionally connectivity (FC) gradients, we closely followed methods outlined by Huntenburg et al.[69] and adapted their code for this purpose (see "Code availability" section). Briefly, *per-run* FC matrices were Fisher r-to-z transformed, averaged across all runs, sessions, and animals, and back-transformed to Pearson's *r* values. The resulting group-averaged FC matrix was decomposed using *diffusion maps*[109], a nonlinear dimensionality reduction method commonly employed by human[67] and mouse[47,69] literature for estimating FC gradients. Gradients have arbitrary units and their absolute value is not meaningful. We z-scored each gradient separately, which allowed us to use a shared color bar in Fig. 8a.

**LFR analysis**
The following parameters need to be specified to generate a binary and overlapping LFR graph[62]. *N*, number of nodes; *k*, average degree; $\mu$, topological mixing parameter; $t_1$, minus exponent for the degree sequence; $t_2$, minus exponent for the community size distribution; $C_{min}$, minimum for the community sizes; $C_{max}$, maximum for the community sizes; *ON*, number of overlapping nodes; *OM*, number of memberships of the overlapping nodes.

To match basic statistics of the real data with LFR graphs we set $N = 542$ (Fig. 1d); and, for every run from each data modality, we calculated the average degree *k* and estimated $t_1$ via an exponential fit to the degree distributions (scipy.stats). We set $t_2 = 0.1$, $C_{min} = 0.05 \times N \approx 27$, $C_{max} = 0.35 \times N \approx 190$. For the fraction of overlapping nodes *ON*, we explored a wide range between 0 (disjoint) and 0.9 in incremental steps of 0.1. This yielded $ON = 0$ up to $ON = 0.9 \times N = 488$. Finally, we used $OM = 2$ and 3. This results in a total of 20 LFR graphs per run, per data modality. We applied the community

detection algorithm to LFR graphs in an identical way to the real data but with fewer seeds ($N = 10$ compared to $N = 500$). The alignment procedure was performed in an identical way as described above, see Supplementary Fig. 7.

**Statistical analysis**
**Hierarchical bootstrapping.** Statistical results were performed at the group level by taking into consideration the hierarchical structure of the data (for each animal, runs within sessions), which can be naturally incorporated into computational bootstrapping to estimate variability respecting the organization of the data[110]. For each iteration (total of 1,000,000), we sampled (with replacement) $n = 10$ animals, $n = 3$ sessions, and $n = 4$ runs, while guaranteeing sessions were yoked to the animal selected and runs were yoked to the session selected (Fig. 1b). In this manner, the multiple runs were always from the same session, which originated from a specific animal. Overall, the procedure allowed us to estimate population-level variability based on the particular sample studied here. To estimate 95% confidence intervals, we used the bias-corrected and accelerated (BCa) method[111], which is particularly effective when relatively small sample sizes are considered (SciPy's scipy.stats.bootstrap). Hierarchical bootstrapping was used to estimate variability in Figs. 2e, f, 3b, 5c, 6c, 7b and Supplementary Figs. 2e, f, 3, 6b, c, 8c, 10.

**One-sample *t*-test.** We used a one-sample *t*-test to define "belonging" in networks as a function of a threshold $\mu$. The *t*-statistic for membership of node $i$ in network $k$ is given as:

$$t_{ik} = \frac{\bar{\pi}_{ik} - \mu}{\text{SE}_{ik}}, \tag{3}$$

where $\bar{\pi}_{ik}$ is the group averaged membership of node $i$ in network $k$, and $\text{SE}_{ik}$ is the standard error estimated using hierarchical bootstrapping (see above). We calculated *p* values using *t*-statistics for all nodes and networks and declared a node a member of a network if its *p* value reached significance $p = 0.05$. The results as a function of various $\mu$ are shown in Fig. 4a. We applied Benjamini–Hochberg correction[112] using Python statsmodels' implementation (statsmodels.stats.multitest.multipletests) to correct for multiple comparisons.

**Permutation test.** Paired permutation tests (two-sided) were used to compare conditions in Fig. 2e, f and Supplementary Figs. 2e, f, 3, 6b, c; and to perform a node-wise comparison across modalities in Fig. 5d. We used SciPy's implementation (scipy.stats.permutation_test) with $N = 1,000,000$ resamples. Holm-Bonferroni correction[113] was applied to correct for multiple comparisons.

**Reporting summary**
Further information on research design is available in the Nature Portfolio Reporting Summary linked to this article.

# Data availability
The raw data analyzed in the present study are available under restricted access because of the complexity of the multimodal data structure and the size of the data. Access can be obtained by contacting the corresponding authors. Data from the Allen Reference Atlas and CCFv3[45] are available on their website: https://portal.brain-map.org/ or through the Allen Software Development Kit (SDK): https://allensdk.readthedocs.io/en/latest/. Source data are provided with this paper.

# Code availability
Our code is available in the following repository: https://github.com/hadivafaii/Ca-fMRI[114]. The open-source software used for overlapping community detection (SVINET[29]) is available in the following

repository: https://github.com/premgopalan/svinet. Our regions of interest definition method (Fig. 1c), makes use of Allen SDK v2.12.3 (https://allensdk.readthedocs.io/en/latest/) and code published by Knox et al.[101] (https://github.com/AllenInstitute/mouse_connectivity_models). The code for generating LFR synthetic graphs[62] can be obtained from https://www.santofortunato.net/resources: "package 1, undirected and unweighted graphs with overlapping communities". To perform functional connectivity gradient analyses, we utilized the code by Huntenburg et al.[69]: https://github.com/juhuntenburg/mouse_gradients. The following software packages were used for data processing: multimodal data registration (BioImage Suite, https://bioimagesuiteweb.github.io/webapp/), fMRI preprocessing (RABIES v0.4.2[35], https://github.com/CoBrALab/RABIES), and Ca$^{2+}$ preprocessing (a pipeline previously published by us[42], https://github.com/YaleMRRC/calPrep). All of our code used for this project is written in Python, making extensive use of Python scientific computing environments, including NumPy v1.20.2[115], SciPy v1.7.0[116], statsmodels v0.13.5[117], scikit-learn v1.2.0[118], pandas v1.3.3[119], matplotlib v3.7.1[120], and seaborn v0.10.1[121].

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

## Acknowledgements

We would like to thank all members of the Multiscale Imaging and Spontaneous Activity in Cortex (MISAC) collaboration at Yale University. We thank P. Brown for valuable input on the design and building of the telecentric lens holder. We thank J. Greenwood and the Neurotechnology Core at Yale University for modifying and building optics associated with the telecentric lens. We thank A. DeSimone, P. Brown, and the Yale School of Medicine electronics and machine shop. This work was supported by funding from NIH R01 MH111424 (R.T.C., M.C., and F.H.), U01 N2094358 (M.C. and R.T.C.), as well as 1RF1NS130069 and R21AG075778 (E.M.R.L.). H.V.'s contribution to this research was supported in part by NSF award DGE-1632976.

## Author contributions

H.V. led the analyses, helped with data registration and preprocessing, and co-wrote the manuscript; F.M. performed multimodal data registrations, data preprocessing, and results interpretation; G.D.G. and D.O. contributed code and helped with data preprocessing; M.M. helped with code testing, analyses, and interpretation; P.H. performed the surgeries; X.S. and X.P. contributed code; M.C. supervised G.D.G. and provided

input on the manuscript; F.H., M.C.C., and R.T.C. supported the collection of the data; X.G. helped with data collection; E.M.R.L. collected the data, helped with analyses, data preprocessing, interpretation of results and edited the manuscript; L.P. supervised the analyses, helped with interpretation of the results, and co-wrote the manuscript; E.M.R.L. and L.P. supervised the study.

## Competing interests

X.P. is a consultant for the Brain Electrophysiology Laboratory Company. X.P. also consults and has an ownership stake in Veridat.

## Additional information

[1]Department of Physics, University of Maryland, College Park, MD 20742, USA. [2]Department of Radiology and Biomedical Imaging, Yale School of Medicine, New Haven, CT 06520, USA. [3]Computional Brain Anatomy Laboratory, Cerebral Imaging Center, Douglas Mental Health University Institute, Montreal, QC H4H 1R3, Canada. [4]Integrated Program in Neuroscience, McGill University, Montreal, QC H3A 0G4, Canada. [5]Department of Biomedical Engineering, Yale University, New Haven, CT 06520, USA. [6]Department of Physiology, School of Medicine, University of California San Francisco, San Francisco, CA 94143, USA. [7]Section of Biomedical Informatics & Data Science, Yale School of Medicine, New Haven, CT 06520, USA. [8]Department of Psychiatry, McGill University, Montreal, QC H3A 0G4, Canada. [9]Department of Biological and Biomedical Engineering, McGill University, Montreal, QC H3A 0G4, Canada. [10]Department of Neuroscience, Yale School of Medicine, New Haven, CT 06510, USA. [11]Kavli Institute for Neuroscience, Yale School of Medicine, New Haven, CT 06510, USA. [12]Department of Ophthalmology and Visual Science, Yale School of Medicine, New Haven, CT 06510, USA. [13]Department of Neurosurgery, Yale School of Medicine, New Haven, CT 06510, USA. [14]Department of Psychology, University of Maryland, College Park, MD 20742, USA. [15]Department of Electrical and Computer Engineering, University of Maryland, College Park, MD 20742, USA. [16]Maryland Neuroimaging Center, University of Maryland, College Park, MD 20742, USA. [17]These authors jointly supervised this work: Evelyn M. R. Lake, Luiz Pessoa. ✉e-mail: vafaii@umd.edu; evelyn.lake@yale.edu; pessoa@umd.edu

