## [Peer Review File · Nature Communications]

REVIEWER COMMENTS

Reviewer #1 (Remarks to the Author):

Comments to the Author:

In this study, Vafai et al. sought to elucidate the similarities and differences between two widely used imaging modalities; fMRI-BOLD and wide-field calcium imaging of genetically-encoded calcium indicators. Their work focused on the overlapping and non-overlapping network structure between imaging modalities and how this was affected by coarse and fine parcellation of the neocortex and number of networks to be identified. As expected, the low-pass filtered calcium signal networks were most similar to the BOLD signal and the slow and fast calcium signal networks were in close agreement. This work provides compelling evidence that BOLD and calcium signals capture similar neuronal dynamics. The methods of investigation used are reasonable and the findings are novel and an important step for comparing imaging modalities which can be extremely difficult.

This work is well designed and investigates the dynamics of the analysis methods used testing multiple numbers of networks and network thresholds. While I find some points of the paper could use revision prior to publication, the work is novel and will add to the field of neuroimaging. I have the following concerns:

- 1) Authors bandpass filter the wide-field calcium signal to closely match the BOLD signal or be somewhat higher frequency than the BOLD signal. It would be interesting and informative to also compare the unfiltered calcium signal to the BOLD signal to see how this changes the results of the network membership.
- 2) In Figure 2C, authors identify seven overlapping communities for each imaging modality. While the spatial regions occupied by the networks are largely similar across imaging techniques, some networks (OC-7) occupy very different spatial regions between modalities. Please provide discussion of why this might happen.
- 3) In relation to the previous point, Figure S1 and S4 show that networks can comprise distant atlas regions yet there is little discussion of how this may come about or what it means for cortical processing. Please discuss.
- 4) Authors show in Figure 3 that OC-2,3, and 7 are some of the most disjointed networks yet little discussion is provided about why this is and what it may mean for processing in the brain. Additionally,

authors threshold the proportion values at $> 20\%$. It would be interesting to see if including these values changes the interpretation of the membership value plots.

5) Pg15Ln6: Authors provide citations with incorrect formatting, please correct this.

6) In Figure 4A, it appears that nodes with multiple network membership are concentrated on the periphery of the overlapping communities. It would be interesting to quantify the spatial extent (rather than just a proportion of nodes) of this multi-network membership to determine if one or more imaging modalities is better at capturing overlap.

7) While authors do a nice job showing mixed network membership and the opposing patterns of degree distribution for the different imaging modalities, there is little discussion about what this might mean for neural processing which is a shame as this could strengthen the discussion significantly.

Reviewer #2 (Remarks to the Author):

In the current study, VaFaii H, et al. utilized a cutting-edge imaging technique combining wide-field Ca^{2+} imaging and 11.7T high-strength fMRI to investigate the patterns of overlapping communities in the intrinsic functional network of the mouse brain. Overall, several research design components (e.g., reproducibility assessment across several parameter changes, the use of an entropy metric, a thorough investigation of both overlapping and disjoint functional networks) as well as methodological advancement are considered as the strengths of the study. Moreover, given that despite enormous field's efforts, the underlying neural origin of spontaneous brain activity in fMRI BOLD signals currently remains unclear, this work may help better understand the implication of intrinsic functional networks derived from fMRI.

Given this importance, however, I have three major concerns, which include the overall impact of the findings, relatively narrow conditions to construct functional networks and a fundamental issue about overlapping-disjoint functional communities.

1. Significance of the findings

1.1 I agree with the authors that the study investigating an overlapping functional organization in the animal field is quite rare, especially those utilizing this kind of state-of-the-art imaging technique. My issue, though, is that simply comparing the patterns of overlapping communities across different

imaging signals may not be the best outcome that can be expected from the simultaneous MR-neural imaging approach. If there were indeed such discrepancies in the network configuration between fMRI and Ca²⁺ imaging, one could anticipate some analyses that can (at least partially) explain where these differences originate from, but the only thing I found in the manuscript was a discussion about a potential difference of the cell origin (excitatory neurons vs cell type agnostic), which was not enough to be conclusive. Otherwise, the existence of overlapping functional networks has been long demonstrated as established organizational principles (e.g., mixed selectivity, compositionality) in the human brain, thus nothing really new (<https://arxiv.org/abs/2209.07431>; <https://www.nature.com/articles/ncomms8751>).

1.2 To address this novelty issue, a simultaneous recording provides unique chances, from which the authors can benefit, for instance, by identifying a set of specific frequency bands from which time-series shows a high correlation with BOLD (not just using Ca²⁺ slow that has a same frequency band than high pass filtered fMRI). In fact, the author's own previous study (Lake et al., 2020, Nat Methods) provided a means to quantify the predictive relationship between Ca²⁺ and BOLD signals using a model that optimizes a Gamma-variant transfer function and found consistent predictions across the cortex, which are best at low frequency (0.009–0.08Hz). This time-series level analysis will tell us how much we could expect Ca²⁺ signals can explain the BOLD-derived metrics and how much we could not.

1.3 The entropy measure from the Ca²⁺ imaging shows boundaries that much more clearly follow the anatomically defined ones from the Allen Institute, compared to the BOLD signals. This was an interesting finding, and I wondered what this difference implies. Is this the indicator for whether BOLD-driven functional communities contain much more artifacts than Ca²⁺ or could reflect some other aspects of functional systems that Ca²⁺ imaging cannot capture? Even in Ca²⁺ imaging, the functional community in the frontal lobe transgresses the anatomical boundaries, and it would be interesting to see what principle makes this community special. Maybe studying underlying structural connectivity could provide hints to address some of these expected and unexpected observations.

2. Generalizability of the findings: One of the limitations of traditional graph-theoretical analyses is that they often relied on a too-specific combination among many different parameter settings. In this regard, the authors successfully demonstrated the robustness of their findings against several major analytical issues (i.e., different thresholds, # of networks, parcellation, and preprocessing pipeline). However, still, the fact that the authors' unit of analysis was binarized connectivity, and that is derived under the (lightly) anesthetized condition, makes the definition of their network construction relatively narrow, and I am not 100% sure how generalizable authors' findings of different community configurations across 3, 7, and 20 networks are towards a true functional organization of the mouse brain. What if the authors did not systematically threshold and just use weighted connectivity strength as they are (maybe only with the noise removal process based on FDR or such), will do the findings still hold? Moreover, although I know it may be overkill (so it is not necessarily something I ask the authors to strictly address), the fact that the focus of this study is an overlapping community and that this functional architecture is likely much more affected during the cognitively awake state, makes me wonder what if they performed exactly same analyses on the awake resting-state data.

3. Continuous functional axes: I think that the authors' overall approach to analyzing the dimension of disjoint-overlapping communities was excellent and scientifically rigorous. I would like to ask them,

though, the opinion about a more extreme perspective towards non-discretized continuous connectome streams, the so-called 'functional gradient'

(www.sciencedirect.com/science/article/pii/S1053811920310132), and how and whether the BOLD and Ca²⁺ imaging would show similar or dissimilar patterns of low-dimensional connectome transition. I think both concepts of 'separable community' and 'continuous functional transition (without clear boundaries)' are two major organizational principles of the brain, so it is a highly relevant issue that can be systematically investigated in the current study.

Reviewer #3 (Remarks to the Author):

Vafaii and colleagues present an investigation of the network organization of the mouse brain using fMRI-BOLD and Ca²⁺ signals. The manuscript details how correlation-based functional connectivity matrices were constructed from these data sources and subsequently analyzed with network methodology. Importantly, the paper applies an overlapping community detection method to investigate how these networks can be partitioned into sets of communities at K=3, 7, and 20. The correspondence between communities derived from each data source is described. The overlapping quality of the communities is measured by taking the entropy of community affiliation at each node and visualized as distributions of membership values. Nodal degree and entropy are compared, to compare two types of overlap scenarios in the data. Overall, the authors present their exploration in a thorough manner, with relevant visualizations and explanation. This manuscript will serve as a fine reference for network neuroscience practitioners seeking to compare community structure in the mouse brain. A drawback of the manuscript is that the analysis employed, in tandem with the data used, is merely descriptive. The manuscript could have had even greater impact and wider interest if experimental task/perturbation data was used to validate the identified communities and to further contrast the organizational structures found in the three data sources. Additionally, it's unfortunate that the community detection algorithm applied, SVINET, requires thresholding and binarization of the functional connectivity matrix, as this is always a hazardous step for network analysis and interpretation. However, I applaud the authors for recognizing this and providing appropriate supplemental analyses to address differences in threshold selection.

Regarding the different data modalities being compared, I would like to have more information about these modalities affect network connectivity characteristics. Specifically, I am interested in how possible differences in the spatial smoothness of the data affect the large difference in regional degree that is shown in figure 6b. Perhaps this comparison could be achieved by plotting the Euclidean distance of edges versus the variance of connectivity values at that distance; this would create a variogram (like here: brainsmash.readthedocs.io/en/latest/approach.html). If the BOLD and Ca²⁺ slow/fast have significantly different variograms, this might contribute to some of the observed differences in topology, and therefore, community structure.

A community detection algorithm, applied with SVINET or modularity maximization or even simple k-means will return a result to you, given that you ask for K clusters. That the SVINET algorithm returns overlapping communities is not surprising to me, given that SVINET has the flexibility to do so. Therefore, as a reader of the manuscript, I do not feel that “significant overlapping structure” is surprising to see. Rather it is what you asked for, by using this algorithm. If you used modularity maximization in combination with other descriptive network tools, you would have found “significant disjoint structure”. The authors might counter that the simulation using the LFR benchmark in figure S5 shows that SVINET correctly identifies disjoint structure in the disjoint case. This is only true in the most extreme case, where planted overlap is indeed, nonexistent, and is not a realistic example of correlation-based functional connectivity matrices anyways. I don’t think the authors actually showed that “considerable overlapping—as opposed to disjoint—network organization was recovered.” My issue here is with the author’s framing of overlapping versus disjoint. I don’t think overlapping structure was compared to disjoint structure directly nor thoroughly, and therefore, I suggest amending the language related to this point. Instead, I think the authors indeed found evidence of an overlapping structure that held across the K parameter and is relevant given the context of previous anatomical and functional neuroanatomy work.

More details and clarifications are needed to describe how results from multiple iterations of SVINET were aggregated to create a centroid community structure. As I understand from the methods, membership vectors were compared via cosine similarity. Are these vectors length 512×1 or $(512 * 7 \text{ communities}) \times 1$ or was it a matrix 512×7 that was compared with some sort of multivariate cosine? It is unclear to me how the fuzzy membership values were handled when aggregating multiple iterations here. And then, after this alignment, were the individual iteration results (i.e. the aligned fuzzy membership data from 500 iterations) averaged for each overlapping community? Relatedly, was the alignment done independently for both Ca²⁺ and BOLD, or were all three aligned to a common set of target centroids?

In Figure 3, what stands out to me is that for many of the distributions, the pattern of overlap proportion seems to make a slight U-shape, with higher values in the 0.2-4, lower in 0.4-0.8, and higher again with 0.8-1.0. Why don’t the authors comment on this pattern, which is not part of the examples illustrated in panel 3a? This pattern seems to indicate that there are many nodes with relatively low overlap and then relatively high evidence for disjointed-ness, but less overlap in the 0.4-0.8 range. This could indicate that when overlap is present, but less 50-50. However, from the data in figure 5a, it does seem like indeed, there are many nodes with 50-50 community affiliation.

It would be helpful if the color bars in Figure 6b were all equal – this way a more straightforward visual comparison could be made. If the authors wish to highlight the similar patterns (despite different magnitudes), I’d recommend visualizing the rank, like what they did in figure 5b.

Author response for manuscript # NCOMMS-23-16500 :

Multimodal measures of spontaneous brain activity reveal both common and divergent patterns of cortical functional organization

Reviewer #1

In this study, Vafaii et al. sought to elucidate the similarities and differences between two widely used imaging modalities; fMRI-BOLD and wide-field calcium imaging of genetically-encoded calcium indicators. Their work focused on the overlapping and non-overlapping network structure between imaging modalities and how this was affected by coarse and fine parcellation of the neocortex and number of networks to be identified. As expected, the low-pass filtered calcium signal networks were most similar to the BOLD signal and the slow and fast calcium signal networks were in close agreement. This work provides compelling evidence that BOLD and calcium signals capture similar neuronal dynamics. The methods of investigation used are reasonable and the findings are novel and an important step for comparing imaging modalities which can be extremely difficult.

This work is well designed and investigates the dynamics of the analysis methods used testing multiple numbers of networks and network thresholds. While I find some points of the paper could use revision prior to publication, the work is novel and will add to the field of neuroimaging. I have the following concerns:

1) Authors bandpass filter the wide-field calcium signal to closely match the BOLD signal or be somewhat higher frequency than the BOLD signal. It would be interesting and informative to also compare the unfiltered calcium signal to the BOLD signal to see how this changes the results of the network membership.

Current best practices for the analyses of our data include bandpass filtering within the ranges we have applied. Since the sources of noise in BOLD-fMRI and wide-field calcium imaging data are different, and the temporal sampling rates are unmatched (1 versus 10 Hz), analyzing these data without the application of a band-pass filter is likely to add sources of inter-modal variance that are not related to the signal

contrasts. Nevertheless, we repeated our analyses without applying frequency filtering, as suggested. The results are summarized below and show that not applying temporal filtering has a very small impact on the results (Figure R1). However, for the reasons mentioned, we do not feel comfortable including the results in the manuscript. Instead, we have added a few sentences to the methods section, as quoted below, to better motivate our application of a frequency filter. Nevertheless, if the reviewer feels strongly about the usefulness of these results, we are open to including them in the supplementary material.

Figure R1: Unfiltered Ca^{2+} data (full frequency range; $[0, 5.0]$ Hz) produces a network structure that is more similar to $\text{Ca}^{2+}_{\text{slow}}$ than $\text{Ca}^{2+}_{\text{fast}}$. This is likely due to the fact that lower frequencies have larger power, which leads them to dominate the results.

The following line was added to Methods (“*RABIES fMRI data preprocessing*”; page 22, lines 560–652):

Current best practices in both human [citation] and mouse [citation] fMRI preprocessing include the application of a high-pass or band-pass filter to remove physiological and other sources of noise. In particular, high-pass filtering removes slow (< 0.01 Hz) drifts from data [citation].

The following was added to Methods (“*Fluorescence Ca^{2+} imaging data preprocessing*”; page 22, lines 580–584):

fMRI-BOLD and wide-field calcium imaging have unmatched temporal sampling rates (1 versus 10 Hz) and different sources of noise. Accordingly, a band-pass filter (3rd order Butterworth) was applied that matched the frequencies of BOLD and $\text{Ca}^{2+}_{\text{slow}}$ ($[0.01, 0.5]$ Hz), and provided a complementary band with higher frequencies ($\text{Ca}^{2+}_{\text{fast}}$: $[0.5, 5.0]$ Hz). The application of band-passing also aids with the removal of noise, including slow drifts.

2) In Figure 2C, authors identify seven overlapping communities for each imaging modality. While the spatial regions occupied by the networks are largely similar across imaging techniques, some networks (OC-7) occupy very different spatial regions between modalities. Please provide discussion of why this might happen.

The points below summarize our view on these issues. Note that the text quoted was part of the original submission (page 18, lines 404–409). If the reviewer believes these points need further elaboration, we are happy to expand on them:

It is possible that some of the discrepancies between BOLD and Ca^{2+} results stemmed from Ca^{2+} signals originating from excitatory neurons, while the BOLD signal is cell-type agnostic. Despite excitatory

neurons being the most populous cell type in the cortex, it is still unclear to what extent the activity of other cell populations such as inhibitory neurons [citation] and astrocytes [citation], or vascular effects [citation], influence the BOLD signal. This will be explored in our future work utilizing the methods established here.

Finally, we note that we also found the differences observed for OC-7 surprising. At present, it is unclear why this network is considerably different between modalities. We now provide a discussion of this point (page 16, lines 331–342). We plan to investigate this issue in detail in the future.

The pronounced differences involving network OC-7 merit particular attention (Figure 2C). In the case of BOLD, this network was centered around parts of the secondary motor cortex known as the frontal orienting field (FOF), considered a homolog of the primate frontal eye field [citations]. For Ca^{2+} signals, FOF was consistently detected as part of a large medial network spanning somatosensory, motor, and parietal cortex (OC-4 in Figure 2C); without forming an independent network (even in the 20-network solution, Figure S1). In contrast, the Ca^{2+} OC-7 network was centered around the retrosplenial area, in a manner that was not captured by any of the seven BOLD networks. We note, however, that the spatially finer 20-network solution for BOLD detected a retrosplenial network, although clearly not to the same extent as identified with Ca^{2+} signals (panel RSP in Figure S1). Overall, we speculate that the differences observed in the case of network OC-7 may reflect particularities of the signal contrasts of the two modalities. To evaluate this question, it will be useful to interrogate Ca^{2+} indicators that are sensitive to different cell populations such as inhibitory neurons and glia (see below).

3) In relation to the previous point, Figure S1 and S4 show that networks can comprise distant atlas regions yet there is little discussion of how this may come about or what it means for cortical processing. Please discuss.

Large-scale networks that include regions that span different lobes are well-established properties of the mammalian brain. Given that these network properties have been extensively discussed in the literature [1, 2], and limited space, we prefer not to reiterate these points in the manuscript. However, if the reviewer and/or editor believe that such a discussion is needed, we are happy to provide further material (one of us even wrote books about it! [see [3]]).

4) Authors show in Figure 3 that OC-2,3, and 7 are some of the most disjoint networks yet little discussion is provided about why this is and what it may mean for processing in the brain.

OCs 2 and 3 cover brain areas that are known for their sensory and motor functionality. However, we were surprised to find that OC-7 was also disjoint for Ca^{2+} data. OC-7 overlaps with the retrosplenial area, a region that participates in multiple functions and is known for its role in multisensory integration. We have added a new paragraph in Discussion (page 17, lines 352–361) that addresses these points:

Measures of network overlap consistently identified the visual (OC-2) and somatomotor (OC-3) networks as among the most disjoint across all conditions. This observation is well aligned with their sensory and motor roles and their potential involvement in fewer processes. Although visual and somatomotor networks (OC-2 and OC-3) exhibited the least amount of overlap among the seven networks, they were still far from being entirely disjoint. This observation is in line with previous proposals that cortical territories should be regarded as essentially multisensory [citation], such that mechanisms of multisensory integration extend into “early” sensory areas (see [citation]). Unexpectedly, in the case of Ca^{2+} data, the retrosplenial area (OC-7) also appeared predominantly as a disjoint network. Given the retrosplenial area’s recognized

multisensory [citation] and multifunctional [citation] characteristics, further investigation is warranted to understand the underlying factors contributing to the organization detected.

Additionally, authors threshold the proportion values at $> 20\%$. It would be interesting to see if including these values changes the interpretation of the membership value plots.

In our simulations, we constructed disjoint networks with no overlap and found that the algorithm detected membership values > 0.8 . In other words, in a completely disjoint network, every node belongs to a single network with > 0.8 strength and, given that a node's membership strength sums to 1, the remaining < 0.2 strength gets distributed across the remaining networks. This establishes a floor value (0.2) for robust network membership. Thus, including values < 0.2 would inadvertently inflate our estimates of overlap (but not alter our conclusion that brain networks have considerable overlap). In sum, given our simulations, we do not believe that it is prudent to consider strengths < 0.2 , and doing so would produce the same take-home messages. To clarify this point, we have reworded the original text (page 6, lines 151–157) where we describe our simulation results:

In simulations, we constructed synthetic disjoint networks with no overlap [citation] and found that the algorithm detected membership values > 0.8 (Figure S7). In other words, in a completely disjoint network, every node belongs to a single network with > 0.8 strength and, given that a node's membership strength sums to 1, the remaining < 0.2 strength is distributed across the remaining networks. This establishes a floor value (0.2) for robust network membership. Thus, to examine membership distributions, we considered the range (0.2, 1.0]; membership values < 0.2 were not considered so as to *conservatively* characterize network overlap.

5) Pg15Ln6: Authors provide citations with incorrect formatting, please correct this.

We thank the reviewer for pointing this out and have corrected the error.

6) In Figure 4A, it appears that nodes with multiple network membership are concentrated on the periphery of the overlapping communities. It would be interesting to quantify the spatial extent (rather than just a proportion of nodes) of this multi-network membership to determine if one or more imaging modalities is better at capturing overlap.

In Figure 4A, we see that regions that belong to two or more networks are distributed across the cortex, but not found in visual or somatomotor areas. However, we don't believe we can determine which modality best captures overlap in our study, given the absence of a clear-cut "ground truth". The contribution of our study is to show that considerable network overlap is detected for the three conditions we investigated (BOLD, Ca_{slow}^{2+} , Ca_{fast}^{2+}).

To further characterize the spatial extent of multi-network membership, we created a figure which separates the four "tiers" of membership (i.e., corresponding to the four thresholds we applied in Figure 4A) Figure R2. What this figure helps visualize is that regions that tend to affiliate with only one network (column 4) are restricted to visual and somatomotor areas. We provide this figure for additional context but do not believe it should be included in the paper because the same information can be gleaned from the original Figure 4.

7) While authors do a nice job showing mixed network membership and the opposing patterns of degree distribution for the different imaging modalities, there is little discussion about what this might mean for

Figure R2: Maps were generated by aggregating regions that belong to each membership *tier*. Size was quantified as the fraction of pixels in each map (values above each map). *Tiers* show a widespread—rather than localized—topography for all thresholds except the most disjoint (column 4, purple). These areas include the sensory and somatomotor (all modalities), and retrosplenial area (Ca²⁺).

neural processing which is a shame as this could strengthen the discussion significantly.

This is an important point and we have considerably expanded the discussion. In brief, we believe the use of degree with BOLD data as a measure of node “importance” is likely problematic (page 17, lines 372–381):

The story became more complicated when we examined region degree, a measure of centrality used at times as an indicator of region “hubness”. Degree showed a negative correlation between BOLD and Ca²⁺ measurements with clearly different spatial patterns. Ca²⁺ degree maps highlighted association areas such as the PTLp, an area that is anatomically and functionally thought to be important for multimodal integration [citation]. In contrast, BOLD degree maps produced counter-intuitive results where, for example, lateral somatomotor areas exhibited high degree though they are likely to be involved in relatively fewer processes. Thus, it appears that while Ca²⁺ degree maps capture region “importance” well, the same cannot be said for BOLD. Indeed, the present finding resonates with previous suggestions that degree centrality alone should not be used to determine the hubness status of brain areas [citation], particularly in the case of BOLD data.

Reviewer #2

In the current study, VaFaii H, et al. utilized a cutting-edge imaging technique combining wide-field Ca²⁺ imaging and 11.7T high-strength fMRI to investigate the patterns of overlapping communities in the intrinsic functional network of the mouse brain. Overall, several research design components (e.g., reproducibility assessment across several parameter changes, the use of an entropy metric, a thorough investigation of both overlapping and disjoint functional networks) as well as methodological advancement are considered as the strengths of the study. Moreover, given that despite enormous field's efforts, the underlying neural origin of spontaneous brain activity in fMRI BOLD signals currently remains unclear, this work may help better understand the implication of intrinsic functional networks derived from fMRI.

Given this importance, however, I have three major concerns, which include the overall impact of the findings, relatively narrow conditions to construct functional networks and a fundamental issue about overlapping-disjoint functional communities.

1. Significance of the findings

1.1 I agree with the authors that the study investigating an overlapping functional organization in the animal field is quite rare, especially those utilizing this kind of state-of-the-art imaging technique. My issue, though, is that simply comparing the patterns of overlapping communities across different imaging signals may not be the best outcome that can be expected from the simultaneous MR-neural imaging approach. If there were indeed such discrepancies in the network configuration between fMRI and Ca²⁺ imaging, one could anticipate some analyses that can (at least partially) explain where these differences originate from, but the only thing I found in the manuscript was a discussion about a potential difference of the cell origin (excitatory neurons vs cell type agnostic), which was not enough to be conclusive. Otherwise, the existence of overlapping functional networks has been long demonstrated as established organizational principles (e.g., mixed selectivity, compositionality) in the human brain, thus nothing really new (<https://arxiv.org/abs/2209.07431>; <https://www.nature.com/articles/ncomms8751>).

Although large-scale brain networks as studied with fMRI are extensively studied in the literature, their correspondence with networks estimated based on other signals remains poorly understood. As the reviewer pointed out, our simultaneous recordings offer multiple potential applications, and establishing the correspondence of large-scale networks is of great importance, we believe. Our approach in doing so was to adopt an overlapping network framework, but we view our contribution as considerably more general: understanding the organization of large-scale networks across signal measures, including their overlapping organization as a function of the frequency band.

Stated more generally, applying simultaneous multimodal measurements to study brain functional organization is an emerging field, and like the reviewer, we have high expectations for the insights that these techniques stand to help the broader community gain. However, our view is a bit different. The origins of the observed discrepancies in network organization *are* due to the different sources of contrast (calcium imaging of excitatory neurons versus the BOLD signal), a conclusion supported by the fact that these data were acquired simultaneously (alongside our efforts to harmonize the data, e.g., application of matched frequency bands). To us, the discovery of both shared and differing network characteristics doesn't necessarily have a simple explanation. As the reviewer points out, overlapping functional organization has been previously observed in humans using BOLD-fMRI data. Here, we are: (1) Demonstrating that this measurement of brain organization extends to mice, where we can (simultaneously) access complementary sources of contrast. (2) Discovering that different sources of contrast (wide-field imaging of excitatory neurons) affirm, or even validate, *some* of what has been observed with BOLD. (3) Identifying aspects

where these modalities converge and diverge. Uncovering the reasons for these inter-modal differences will require further investigation. For example, simultaneous imaging of both excitatory and inhibitory neural activity alongside the BOLD-fMRI signal is something we are currently developing.

1.2 To address this novelty issue, a simultaneous recording provides unique chances, from which the authors can benefit, for instance, by identifying a set of specific frequency bands from which time-series shows a high correlation with BOLD (not just using Ca²⁺ slow that has a same frequency band than high pass filtered fMRI). In fact, the author's own previous study (Lake et al., 2020, Nat Methods) provided a means to quantify the predictive relationship between Ca²⁺ and BOLD signals using a model that optimizes a Gamma-variant transfer function and found consistent predictions across the cortex, which are best at low frequency (0.009–0.08Hz). This time-series level analysis will tell us how much we could expect Ca²⁺ signals can explain the BOLD-derived metrics and how much we could not.

To address the suggestion of the reviewer, we repeated our network analyses by applying an HRF filter to calcium signals, which we obtained from our previous work. The results are shown in a new Figure S6. Overall, filtering calcium signals changed network organization in relatively modest ways. Nevertheless, a quantitative analysis of the differences did reveal some that were statistically significant. We present these findings and discuss them briefly on page 6, lines 133–138:

To further probe some of the differences between BOLD and Ca²⁺ results, we repeated our network similarity analysis after applying a gamma-variate hemodynamic filter [citation] to the calcium signal, which we obtained from our previous work [citation] (Figure S6). By doing so, Ca²⁺ data would presumably better approximate BOLD data. Filtering Ca²⁺ signals changed network organization in relatively modest ways for Ca²⁺_{slow}, although some of the differences were statistically significant (Figure S6B). Filtering produced more pronounced quantitative changes to Ca²⁺_{fast} networks, especially OC-4, OC-5, and OC-6.

1.3 The entropy measure from the Ca²⁺ imaging shows boundaries that much more clearly follow the anatomically defined ones from the Allen Institute, compared to the BOLD signals. This was an interesting finding, and I wondered what this difference implies. Is this the indicator for whether BOLD-driven functional communities contain much more artifacts than Ca²⁺ or could reflect some other aspects of functional systems that Ca²⁺ imaging cannot capture? Even in Ca²⁺ imaging, the functional community in the frontal lobe transgresses the anatomical boundaries, and it would be interesting to see what principle makes this community special. Maybe studying underlying structural connectivity could provide hints to address some of these expected and unexpected observations.

The reviewer brings up a fascinating point. Interestingly, our own view seems to differ considerably from that of the reviewer, as we do not expect functional and anatomical boundaries to correspond with each other. Our view is that in complex systems like the brain, whereas anatomy clearly provides important constraints to function, the mapping between the two is far from straightforward. In fact, we believe this is one of the fundamental problems in systems neuroscience and have discussed it at length in other publications. Our goal is to investigate these very issues in future work.

2. Generalizability of the findings: One of the limitations of traditional graph-theoretical analyses is that they often relied on a too-specific combination among many different parameter settings. In this regard, the authors successfully demonstrated the robustness of their findings against several major analytical issues (i.e., different thresholds, # of networks, parcellation, and preprocessing pipeline). However, still, the fact that the authors' unit of analysis was binarized connectivity, and that is derived under the (lightly) anesthetized condition, makes the definition of their network construction relatively narrow, and I am not

100% sure how generalizable authors' findings of different community configurations across 3, 7, and 20 networks are towards a true functional organization of the mouse brain. What if the authors did not systematically threshold and just use weighted connectivity strength as they are (maybe only with the noise removal process based on FDR or such), will do the findings still hold? Moreover, although I know it may be overkill (so it is not necessarily something I ask the authors to strictly address), the fact that the focus of this study is an overlapping community and that this functional architecture is likely much more affected during the cognitively awake state, makes me wonder what if they performed exactly same analyses on the awake resting-state data.

To address the generalizability of our findings, we performed initial, exploratory analysis in a group of $N = 5$ animals for which we had Ca^{2+} recordings in both anesthetized and awake states. These initial results are extremely encouraging and are now covered in the Discussion (page 18, lines 418–425), and include new supplementary Figure S5:

Nevertheless, to evaluate the impact of anesthesia on our results, we performed initial, exploratory analyses in a group of $N = 5$ animals for which we had Ca^{2+} recordings in both anesthetized and awake states. These animals, which participated in the sessions reported in the main results, underwent an additional Ca^{2+} recording session outside of the scanner while awake. The results (Figure S5) clearly show that the large-scale network organization that we reported in the anesthetized state is also observed in awake mice. Relatively minor changes are also observed but require a larger sample for proper statistical comparisons. Nevertheless, the findings with this small group of animals considerably strengthen the generalizability of our findings.

We also added the following sentence to the Results section “*Cortical organization captured by overlapping network solutions*” (page 5, lines 110–113):

We also investigated the 7-network organization in a subset of animals that underwent an additional awake imaging session that measured Ca^{2+} signals outside the MRI scanner. Notably, the overall organization in awake animals (Figure S5) was qualitatively very similar to that obtained with lightly anesthetized animals.

Finally, we note that we did not perform the analysis suggested by the reviewer with unthresholded functional connectivity matrices because the Bayesian generative algorithm we employed to estimate overlapping networks requires binary matrices. Developing an extension of the algorithm for weighted matrices is beyond the scope of the present study. However, we note that the analysis of functional gradients added to the main Results does utilize unthresholded correlation matrices (see next point).

3. Continuous functional axes: I think that the authors' overall approach to analyzing the dimension of disjoint-overlapping communities was excellent and scientifically rigorous. I would like to ask them, though, the opinion about a more extreme perspective towards non-discretized continuous connectome streams, the so-called ‘functional gradient’ (www.sciencedirect.com/science/article/pii/S1053811920310132), and how and whether the BOLD and Ca^{2+} imaging would show similar or dissimilar patterns of low-dimensional connectome transition. I think both concepts of ‘separable community’ and ‘continuous functional transition (without clear boundaries)’ are two major organizational principles of the brain, so it is a highly relevant issue that can be systematically investigated in the current study.

We would like to thank the reviewer for this excellent suggestion, which we had not considered. Indeed, the emphasis on the continuous nature of brain organization of the gradient framework resonates well with our overlapping community approach. Following the suggestion, we analyzed our data in terms of functional gradients, which are now presented in two new Results sections, “*Nearly identical principal*

functional gradient across modalities” and “*Functional gradients of the cortex*”, which discuss the new Figure 8. We also discuss the implications of these new findings in the Discussion (page 18, lines 391–403):

Functional gradients characterize continuous axes of variation and/or organization across the cortex [citation]. Here, we investigated the top four gradients, which revealed that the principal one was nearly identical across modalities, and captured a visual-to-somatomotor axis (Figure 8). Indeed, other studies have identified this gradient as a key organizational feature of the mouse cortex [citation]. Two other gradients (G-2 and G-3) were also well captured across modalities, although their ordering (based on eigenvalues) was different (but note that the percentage of variance explained was very similar). Finally, the spatial organization of G-4 was not reflected in the gradients identified based on Ca^{2+} signals. Intriguingly, this gradient highlights regions overlapping with the frontal orienting field captured by BOLD OC-7 (Figure 2C), a network that was not detected by Ca^{2+} even in the case of 20-network decomposition (Figure S1). It is also noteworthy that some of the gradients detected spatial organization that was not reflected in the community maps. Overall, our results unveiled three functional gradients that exhibited strong to intermediate correspondence across conditions, as well as one spatial organization seen in BOLD that was not present in Ca^{2+} signals.

Reviewer #3

Vafaii and colleagues present an investigation of the network organization of the mouse brain using fMRI-BOLD and Ca^{2+} signals. The manuscript details how correlation-based functional connectivity matrices were constructed from these data sources and subsequently analyzed with network methodology. Importantly, the paper applies an overlapping community detection method to investigate how these networks can be partitioned into sets of communities at $K=3, 7, \text{ and } 20$. The correspondence between communities derived from each data source is described. The overlapping quality of the communities is measured by taking the entropy of community affiliation at each node and visualized as distributions of membership values. Nodal degree and entropy are compared, to compare two types of overlap scenarios in the data. Overall, the authors present their exploration in a thorough manner, with relevant visualizations and explanation. This manuscript will serve as a fine reference for network neuroscience practitioners seeking to compare community structure in the mouse brain. A drawback of the manuscript is that the analysis employed, in tandem with the data used, is merely descriptive. The manuscript could have had even greater impact and wider interest if experimental task/perturbation data was used to validate the identified communities and to further contrast the organizational structures found in the three data sources. Additionally, it's unfortunate that the community detection algorithm applied, SVINET, requires thresholding and binarization of the functional connectivity matrix, as this is always a hazardous step for network analysis and interpretation. However, I applaud the authors for recognizing this and providing appropriate supplemental analyses to address differences in threshold selection.

We agree with the reviewer that having to binarize the functional connectivity matrices was unfortunate, and plan to circumvent this issue in the future by developing novel algorithms that do not require such steps. With that said, we have now included new results on functional connectivity gradients that are computed directly from the weighted values and do not require thresholding. These results are presented as a new Figure 8, two new Results sections, “*Nearly identical principal functional gradient across modalities*” and “*Functional gradients of the cortex*”, and further discussed in the Discussion (page 18, lines 391–403).

Regarding the different data modalities being compared, I would like to have more information about these modalities affect network connectivity characteristics. Specifically, I am interested in how possible differences in the spatial smoothness of the data affect the large difference in regional degree that is shown in figure 6b. Perhaps this comparison could be achieved by plotting the Euclidean distance of edges versus the variance of connectivity values at that distance; this would create a variogram (like here: brainsmash.readthedocs.io/en/latest/approach.html). If the BOLD and Ca²⁺ slow/fast have significantly different variograms, this might contribute to some of the observed differences in topology, and therefore, community structure.

We agree that different levels of data smoothness can potentially contribute to some of the observed differences between the modalities studied. As the reviewer’s point focused on the degree maps (Figure 6), we evaluated the impact of spatial smoothing on such maps. To do so, we applied a Gaussian kernel to smoothen Ca_{slow}²⁺ data, followed by computing degree maps using steps that were otherwise identical to those used originally. We explored various degrees of smoothness quantified by the full width at half maximum (FWHM) of the filter (indicated on top of the maps in Figure R3 further below). Irrespective of the amount of smoothing of the calcium signals, the resulting degree maps were spatially very similar to that of the original degree maps, and substantially different from the BOLD degree map. It is worth noting that Ca_{slow}²⁺ smoothing did produce degree maps that were substantially smoother than the original one, but which were nevertheless quite different from the BOLD spatial organization. These results provide strong evidence that the reason the degree map for Ca_{slow}²⁺ is qualitatively different from that of the BOLD degree map is not due to data smoothness, and instead reflects differences in the underlying contrast signals themselves (i.e., BOLD vs. Ca²⁺).

Figure R3: Ca²⁺ degree maps at different levels of data smoothness. To obtain these results, we applied a Gaussian filter to raw Ca²⁺ data followed by otherwise identical steps in our pipeline, including Ca_{slow}²⁺ bandpassing. A considerable amount of data smoothing (full width at half maximum = 1.9 mm) results in a Ca_{slow}²⁺ degree map that is comparable to that of BOLD in terms of smoothness; however, the spatial patterns remain substantially different across modalities.

Furthermore, we now discuss potential shortcomings of using BOLD degree maps to capture region “importance” or “hubness”. This is now discussed in the Discussion (page 17, lines 372–381), which also addressed point 7 of Reviewer 1:

The story became more complicated when we examined region degree, a measure of centrality used at times as an indicator of region “hubness”. Degree showed a negative correlation between BOLD and Ca²⁺ measurements with clearly different spatial patterns. Ca²⁺ degree maps highlighted association areas such as the PTLp, an area that is anatomically and functionally thought to be important for multimodal integration [citation]. In contrast, BOLD degree maps produced counter-intuitive results where, for example, lateral somatomotor areas exhibited high degree though they are likely to be involved in relatively fewer processes. Thus, it appears that while Ca²⁺ degree maps capture region “importance” well, the same cannot be said for BOLD. Indeed, the present finding resonates with previous suggestions that degree centrality

alone should not be used to determine the hubness status of brain areas [citation], particularly in the case of BOLD data.

A community detection algorithm, applied with SVINET or modularity maximization or even simple k-means will return a result to you, given that you ask for K clusters. That the SVINET algorithm returns overlapping communities is not surprising to me, given that SVINET has the flexibility to do so. Therefore, as a reader of the manuscript, I do not feel that “significant overlapping structure” is surprising to see. Rather it is what you asked for, by using this algorithm. If you used modularity maximization in combination with other descriptive network tools, you would have found “significant disjoint structure”. The authors might counter that the simulation using the LFR benchmark in figure S5 shows that SVINET correctly identifies disjoint structure in the disjoint case. This is only true in the most extreme case, where planted overlap is indeed, nonexistent, and is not a realistic example of correlation-based functional connectivity matrices anyways. I don’t think the authors actually showed that “considerable overlapping—as opposed to disjoint—network organization was recovered.” My issue here is with the author’s framing of overlapping versus disjoint. I don’t think overlapping structure was compared to disjoint structure directly nor thoroughly, and therefore, I suggest amending the language related to this point. Instead, I think the authors indeed found evidence of an overlapping structure that held across the K parameter and is relevant given the context of previous anatomical and functional neuroanatomy work.

We have revised the manuscript to emphasize both disjoint and overlapping organization, thus providing a more balanced account. To clarify, given that the vast majority of the literature on large-scale networks emphasizes a disjoint organization, we sought to test for evidence of non-trivial overlapping organization. In that respect, the language might have emphasized overlapping, but of course, the organization characterized in our study is both disjoint and overlapping. We added the following sentences to Discussion, page 441, lines 441–445:

Finally, we would like to emphasize that while we believe our results provide evidence for considerable network overlap, estimating the extent of overlap more quantitatively remains a challenging problem in network science. Important progress in this direction involves developing principled approaches based on statistical inference and generative modeling [citation].

More details and clarifications are needed to describe how results from multiple iterations of SVINET were aggregated to create a centroid community structure. As I understand from the methods, membership vectors were compared via cosine similarity. Are these vectors length 512×1 or $(512 * 7 \text{ communities}) \times 1$ or was it a matrix 512×7 that was compared with some sort of multivariate cosine?

Each membership vector has shape 512×1 . To align them, we always performed pairwise comparisons, using regular cosine similarity. Specifically, to align a set of K membership vectors (source) to another set of K membership vectors (target), we perform cosine similarity on all pairs yielding a $K \times K$ similarity matrix from source to target. We then reorganize rows such that diagonals have maximum average similarity, a procedure known as the Hungarian algorithm. We have added these clarifications to Methods and have quoted them below (please see highlighted text below).

It is unclear to me how the fuzzy membership values were handled when aggregating multiple iterations here. And then, after this alignment, were the individual iteration results (i.e. the aligned fuzzy membership data from 500 iterations) averaged for each overlapping community?

To obtain the result across iterations, we averaged over individual iterations after the alignment was done (please see highlighted text below).

Relatedly, was the alignment done independently for both Ca²⁺ and BOLD, or were all three aligned to a common set of target centroids?

The alignment was done separately for BOLD, Ca_{slow}²⁺, and Ca_{fast}²⁺. We have now revised the related Methods section (“*Aligning community results*”; page 24, lines 640–650) and we are copying the revised text below for the reviewer’s consideration:

Communities were identified in random order due to the stochastic nature of our algorithm. Maximum cosine similarity of the cluster centroids was used to match communities across calculations (runs or random seeds), as follows. For each run, membership vectors from all random seeds were submitted to *k*-means clustering (sklearn.cluster.KMeans) to determine *K* clusters (e.g., *K* = 7 for analyses with 7 communities). The similarity between membership vectors from each random seed (source) and the cluster centroids (target) was then established via pairwise cosine similarity, yielding a *K* × *K* similarity matrix from source to target per random seed. A permutation of the rows of this similarity matrix was identified such that diagonals had maximum average similarity, a procedure known as the Hungarian algorithm. Finally, the identified permutation was applied to seed results, thereby aligning them with the targets. The aligned communities were then averaged. The outcome was membership matrix π (Figure 1E). Importantly, this matching procedure was done for each condition (BOLD, Ca_{slow}²⁺, Ca_{fast}²⁺) separately.

In Figure 3, what stands out to me is that for many of the distributions, the pattern of overlap proportion seems to make a slight U-shape, with higher values in the 0.2-4, lower in 0.4-0.8, and higher again with 0.8-1.0. Why don’t the authors comment on this pattern, which is not part of the examples illustrated in panel 3a? This pattern seems to indicate that there are many nodes with relatively low overlap and then relatively high evidence for disjointed-ness, but less overlap in the 0.4-0.8 range. This could indicate that when overlap is present, but less 50-50. However, from the data in figure 5a, it does seem like indeed, there are many nodes with 50-50 community affiliation.

This is a good point; we failed to discuss this point more clearly, and have now added the following text to Results (“*Cortical networks show prominent overlapping organization*”; page 6, lines 164–171):

For several of the conditions, we observed a roughly reverse L-shaped distribution. In the case of OC-2, OC-3, and OC-7 they were reminiscent of the disjoint pattern of Figure 3A, except that the “base” has a considerably higher level than zero. Thus, these networks have considerable disjoint organization, in some cases with 60% of the regions affiliating with a single network. OC-1, OC-5, and OC-6 exhibited a more true U shape with relatively more strongly affiliated regions (right side) and weakly affiliated regions (left side), with relatively fewer regions with intermediate membership strengths (middle two values). Nevertheless, it is important to note that in many cases the proportion of regions with intermediate membership values (0.4–0.8 range) was around 40%.

It would be helpful if the color bars in Figure 6b were all equal – this way a more straightforward visual comparison could be made. If the authors wish to highlight the similar patterns (despite different magnitudes), I’d recommend visualizing the rank, like what they did in figure 5b.

We agree, and to enhance the comparability of degree maps, we replaced Figure 6b to use degree ranks. We included a supplementary Figure S9 where we show the actual degree values, but now with a common color bar across modalities.

References

- [1] Olaf Sporns. *Networks of the Brain*. MIT press, 2016.
- [2] Alex Fornito, Andrew Zalesky, and Edward Bullmore. *Fundamentals of brain network analysis*. Academic press, 2016.
- [3] Luiz Pessoa. *The entangled brain: How perception, cognition, and emotion are woven together*. MIT Press, 2022.

REVIEWERS' COMMENTS

Reviewer #1 (Remarks to the Author):

Review – Multimodal measures of spontaneous brain activity reveal both common and divergent patterns of cortical functional organization

Comments to the Author:

In this study, Vafaii et al. provide a close comparison fMRI-BOLD and wide-field calcium imaging data in the same animals. This work reveals considerable cross-modality network overlap as well as some non-overlap. While the BOLD signals and calcium signal are not strictly identical, the authors do a comprehensive job in characterizing the networks between the modalities. This work provides compelling evidence that BOLD and calcium signals capture similar neuronal dynamics. The methods of investigation used are reasonable and the findings are novel and an important step for comparing imaging modalities. This work investigates the dynamics of the analysis methods used testing multiple numbers of networks and network thresholds.

I find that upon revision and secondary review, the authors took care to address all reviewer concerns (including my own). When necessary, authors provided an expanded explanation for methodological choices. Additionally authors, to the extent reasonable and feasible, provided additional proof of concept and supporting analyses to further strengthen their findings. In conclusion, the findings of the study are novel and critical to comparing imaging modalities and I would recommend this paper for publication.

Reviewer #2 (Remarks to the Author):

I would like to thank the authors' efforts to address my comments. I went through all their responses carefully and found that the level of evidence is now increased by adding new results such as the demonstration of awake condition and gradient analyses.

I have no further issues.

Reviewer #3 (Remarks to the Author):

The authors appropriately addressed the comments and concerns that I previously raised. Good job.